# A biobank of pediatric patient-derived-xenograft models in cancer precision medicine trial MAPPYACTS for relapsed and refractory tumors

Maria Eugénia Marques Da Costa [1,2], Sakina Zaidi [3], Jean-Yves Scoazec [4], Robin Droit[1,5], Wan Ching Lim[1,6], Antonin Marchais[1,2], Jerome Salmon[1], Sarah Cherkaoui[1,7], Raphael J. Morscher[1,7], Anouchka Laurent[8], Sébastien Malinge [8,9], Thomas Mercher[8], Séverine Tabone-Eglinger [10], Isabelle Goddard[11], Francoise Pflumio [12], Julien Calvo [12], Francoise Redini[13], Natacha Entz-Werlé[14], Aroa Soriano[15], Alberto Villanueva [16], Stefano Cairo[17], Pascal Chastagner[18], Massimo Moro [19], Cormac Owens[20], Michela Casanova[19], Raquel Hladun-Alvaro[15], Pablo Berlanga[2], Estelle Daudigeos-Dubus [1], Philippe Dessen[1,5], Laurence Zitvogel [1], Ludovic Lacroix[4], Gaelle Pierron [21], Olivier Delattre [3,21,22], Gudrun Schleiermacher [3,22], Didier Surdez [3,23,24] & Birgit Geoerger [1,2,24✉]

Pediatric patients with recurrent and refractory cancers are in most need for new treatments. This study developed patient-derived-xenograft (PDX) models within the European MAP-PYACTS cancer precision medicine trial (NCT02613962). To date, 131 PDX models were established following heterotopical and/or orthotopical implantation in immunocompromised mice: 76 sarcomas, 25 other solid tumors, 12 central nervous system tumors, 15 acute leukemias, and 3 lymphomas. PDX establishment rate was 43%. Histology, whole exome and RNA sequencing revealed a high concordance with the primary patient's tumor profile, human leukocyte-antigen characteristics and specific metabolic pathway signatures. A detailed patient molecular characterization, including specific mutations prioritized in the clinical molecular tumor boards are provided. Ninety models were shared with the IMI2 ITCC Pediatric Preclinical Proof-of-concept Platform (IMI2 ITCC-P4) for further exploitation. This PDX biobank of unique recurrent childhood cancers provides an essential support for basic and translational research and treatments development in advanced pediatric malignancies.

---

*A list of author affiliations appears at the end of the paper.

Internationally standardized multimodal treatments now achieve a 5-year overall survival above 85% in children, adolescents and young adults with cancer[1]. This figure however varies greatly between tumor types and despite highly toxic therapy, nearly half of cancers escape current first line treatment either by primary resistance or relapse after initial response. "MoleculAr Profiling for Pediatric and Young Adult Cancer Treatment Stratification" (MAPPYACTS; ClinicalTrials.gov identifier: NCT02613962) is an international, multicenter, prospective cancer precision medicine trial with the main objective to match relapsed or refractory pediatric patients to treatment with targeted agents based on individual advanced molecular tumor profiles[2]. A central ancillary study of the trial was the establishment of relevant preclinical patient-derived xenograft (PDX) models with deep phenotype characterization provided here.

PDX models, directly xenotransplanted from the primary patient tumor, are now widely in use as high impact preclinical models. They have been shown to retain tumor heterogeneity, histological characteristics, genomic profiles, as well as response patterns to therapies[3–12]. Despite the limitations that come with every model system, they have advanced our understanding in large cohorts of adult[3,13] as well as pediatric and adolescent cancer types[4,7,14,15]. A small fraction of reports highlights genomic discordances, with large chromosomal rearrangements or gene amplifications and deletions acquired in pediatric PDX models[16]. Deep phenotyping of the PDX models is therefore imperative and sets the basis for providing a high impact resource to the community. This is especially relevant, as we provide a sample-matched patient cohort with available deep molecular phenotyping and detailed clinical data.

Knowing that spatial heterogeneity and temporal evolution are key factors in resistance development[17,18], recent cancer precision medicine trials prioritize biopsies after progression or relapse and characterize tumors to identify newly acquired vulnerabilities. This study specifically serves the need for matched models at this treatment stage and provide a resource at relapsed/refractory stage, as opposed to PDX libraries derived from tumors at diagnosis. Within MAPPYACTS, we have generated a collection of 131 well-characterized pediatric PDX models in a comprehensive effort of clinical and laboratory research centers in France, Ireland, Italy, and Spain.

## Results

**Patient characteristics and PDX establishment.** Between February 2016 and July 2020, 787 pediatric, adolescent and young adult patients with recurrent or refractory malignancies were enrolled in the MAPPYACTS trial;[2] 756 (96%) patients and their parents consented to the optional ancillary study of preclinical model development (Fig. 1a). 744 patients had a cancer tissue sampling procedure performed; 12 patients were screening failures. Median age at time of inclusion was 11.9 years (range, 0.5–38.5), 59% were male. All patients had received anticancer treatment before study procedure, aligned with the inclusion criteria of MAPPYACTS. All had prior systemic chemotherapy regimens, except two patients who had radiation therapy only; 31 PDX were derived from tumor lesions that had been irradiated. Fifty patients had a second and five patients had a third sample acquisition, in most cases because of insufficient tumor sample or unsuccessful sequencing; six patients had two successful sequencing analyses at different time points of their disease.

Figure 1, Supplementary Fig. S1 and Supplementary Table S1 depict the workflow and patient flowchart, samples processed as well as all PDX models established, respectively. From 799 procedures in 744 patients, 505 cancer samples (63%) were received in 10 research laboratories for PDX development. To date, 302 samples were transplanted directly into mice, 147 engrafted successfully (49%) in the first passage (P0) and 131 models were considered as stable/established (i.e., reached the 2nd passage in vivo (P2) in 123 PDX and eight leukemia considered established at first tumor take) (Fig. 1a–c). A total of 223 patient tumor samples have been soft-frozen in laboratories, including from 209 patients for whom no model could be transplanted or established (Supplementary Table S1).

Primary tumor take rates and tumor model establishment rates varied for the malignancies between 0% and 82% (Fig.1b and Supplementary Table S1). Figure 1c displays the 131 established PDX models per histology and the laboratory that developed it.

**Characteristics of the established PDX models.** One hundred thirty-one PDX obtained from patients with 0.5–30.8 years old were considered as stable and established models, 113 were solid tumors (76 sarcoma, 25 other non-central nervous system (CNS) tumors, 12 CNS tumors), 3 lymphomas and 15 leukemias presenting with more than 50% leukemic blasts. Supplementary Data 1 details the characteristics of all 131 PDX models and their originating patient tumor with origin, prior therapies, molecular alterations retained in the clinical molecular tumor board that were considered as actionable, of interest or disease specific, IMI2 ITCC-P4 identifiers and other model names. Established PDX models underwent further characterization for histology, WES, RNA-Seq, HLA typing and growth behavior. To date, 90 of the 131 solid tumor MAPPYACTS PDX models were shared with the European Innovative Medicines Initiative 2 (IMI2) ITCC-P4 (Innovative Therapies for Children with Cancer-Pediatric Preclinical Proof-of-concept Platform) initiative (Project Number: 116064; IMI2/INT/2015-03842 v.2020; https://www.itccp4.eu/), where they will be used for drug screening and exploration.

**Growth characteristics of established PDX.** The in vivo characteristics at P2 passage for 70 PDX models were grouped by tumor type (Fig. 2 and Supplementary Fig. S2). Tumor take of the first passage (P0) ranged between 12 and 285 days (median, 41 days), while growth accelerated in general for all models during P1 and P2 to a median of 21 days (range, 5 and 104).

**Morphologic primary tumor type is preserved in established PDX.** At each passage, hematoxillin-eosin-safranin (HES) staining was performed on PDX tissue sections for morphological analysis of concordance with the primary tumor. Fifty-one pairs of primary tumors versus PDX were evaluated centrally in one laboratory by one pathologist with expert experience in pediatric tumors (JY Scoazec). In a first step, representative sections of PDX tumors were examined without knowledge of the initial patient diagnosis and a tentative diagnosis based on morphology alone was proposed. In a second step, representative sections of both primary tumors and their corresponding PDX were compared to classify their respective histological features as similar (morphology fully preserved, including some distinctive features of the primary), comparable (morphology slightly divergent but still suggestive, loss of distinctive features of the primary) or different. In 39 cases, the PDX-based diagnosis was concordant with the initial diagnosis; histological features were similar in 29 cases and comparable in 10. In 12 cases, PDX-based diagnosis was discordant; in all cases, the initial diagnosis was confirmed after review of histological, immunohistochemical and/or molecular data. Three causes of discrepancies were identified: (1) loss of distinctive histological features resulting in a morphologically undifferentiated tumor (n = 5), even if additional studies confirmed the preservation of diagnostic immunohistochemical markers such as MyoD1 in rhabdomyosarcoma; (2) expansion of

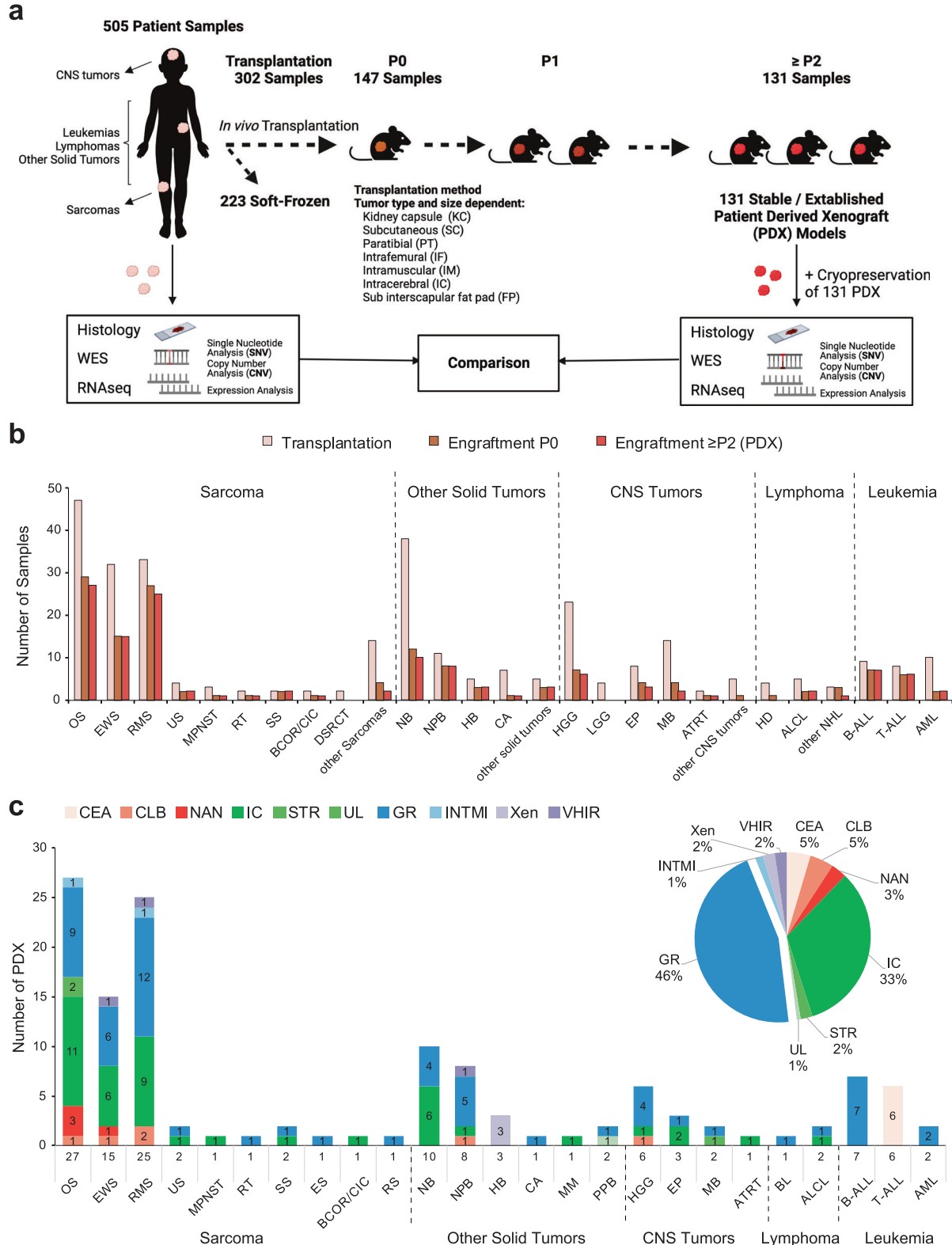

a blastic component without preservation of differentiated features ($n = 3$); (3) true diagnostic errors, mainly in cases in which morphology alone was not sufficient to reach a final diagnosis ($n = 4$).

Representative examples of the primary tumor and at stable passage (P2) are presented in Fig. 3 and detailed results of the comparative study are given in Supplementary Table S2. In these 51 PDX tumors, necrosis was usually absent ($n = 48/51$)

or focal ($n = 3/51$), which confirms the quality and sustainability of their growth. Mouse stromal areas ranged from 1% to maximum 61% (mean 13.8%) of the PDX sample at P2 as defined by a *TBP* murine and human specific RT-qPCR assay in 38 PDX; osteosarcoma PDX at P2 had significantly more ($p = 0.046$) murine stroma (mean $24.2 \pm 18.9\%$ standard deviation) as compared to the other PDX entities ($10.2 \pm 8.6\%$).

**Fig. 1 Workflow for PDX development in MAPPYACTS and established models. a** 2–5 mm$^3$ solid tumor fragments were collected from resection or biopsy samples and transplanted (subcutaneously (SC) or orthotopically, see Supplementary Table S1 and Supplementary Data 1 for details) in immunocompromised mice (NSG, NSG-IL, SCID-beige, Swiss nude). For central nervous system (CNS) tumors, samples were digested and cell homogenates were injected subcutaneously and/or intracerebrally (IC) into the caudate nucleus, cerebellum or pons of nude mice. To generate leukemia PDX, 0.05–1 × 10$^6$ cells from bone marrow or peripheral blood were injected into sub-lethally irradiated (2.5 Gy) NSG mice by intrafemoral injection. Tumor take was monitored during 6–18 months or 1 year for leukemias. Model characterization was performed using morphological and molecular analyses from passages (P1 to P4) and validated by comparing it to the patient's tumor. **b** Plot represents numbers of tumor samples transplanted (left bars), samples with first tumor take in P0 (middle bars) and established models in ≥P2 or P1 in some leukemias (right bars). **c** Graph shows numbers of established models per histological tumor type and the research center in which they were established. B-ALL/T-ALL acute lymphoblastic leukemia, AML acute myeloid leukemia, ALCL anaplastic large cell lymphoma, HD Hodgkin lymphoma, BL Burkitt lymphoma, NHL non-Hodgkin lymphoma, OS osteosarcoma, EWS Ewing sarcoma, RMS rhabdomyosarcoma, RT rhabdoid tumor, BCOR/CIC BCOR or CIC-translocated sarcoma, SS synovial sarcoma, MPNST malignant peripheral nerve sheath tumor, DSRCT desmoplastic small round cell tumor, ES epithelioid sarcoma, US undifferentiated sarcoma, RS renal sarcoma, NB neuroblastoma, NPB nephroblastoma, CA carcinoma, HB hepatoblastoma, PPB pleuropulmonary blastoma, MM melanoma, HGG high-grade glioma, LGG low grade glioma, MB medulloblastoma, EP ependymoma, ATRT Atypical teratoid rhabdoid tumor, CEA Commissariat à l'énergie atomique et aux énergies alternatives, CLB Centre Léon Bérard, NAN CHU Nantes, IC Institut Curie, STR CHU Strasbourg, UL Nancy Lorraine University, GR Gustave Roussy, XEN Xentech, VHIR Vall d'Hebron Research Institute, INTMI Fondazione IRCCS Istituto Nazionale dei Tumori, Milan, Italy.

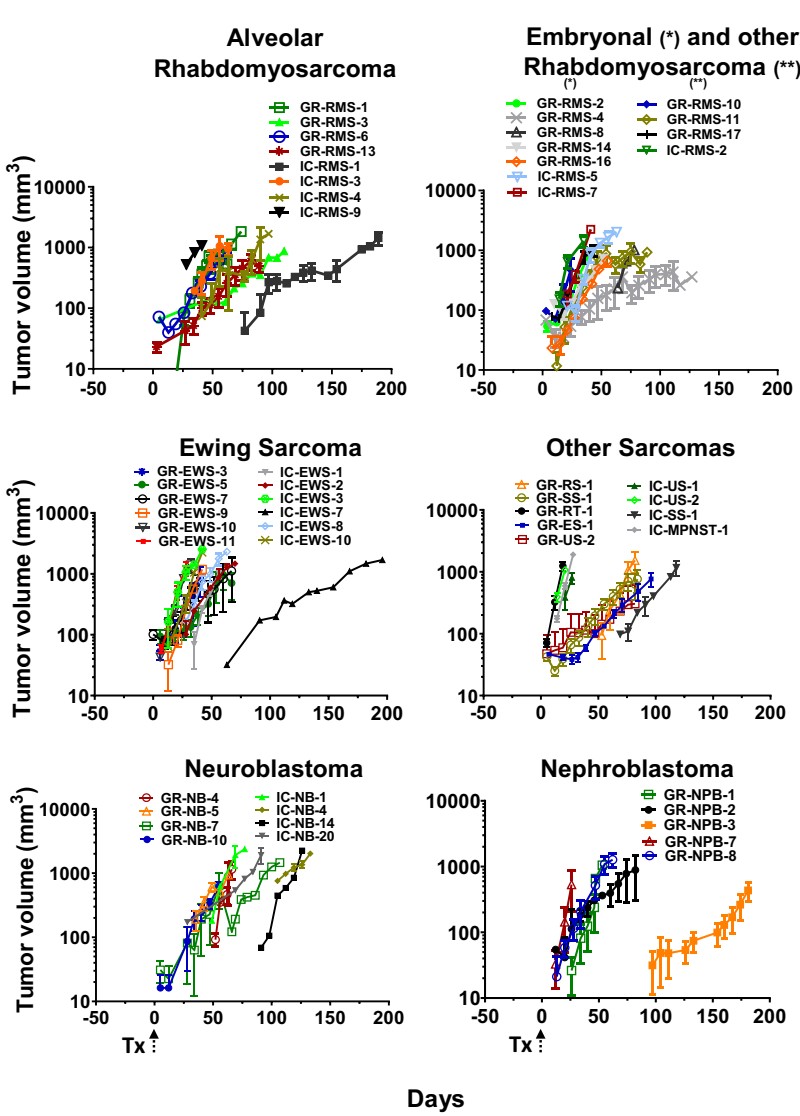

**Fig. 2 PDX tumor growth.** Tumor growth of established subcutaneous alveolar, embryonal or other rhabdomyosarcoma, Ewing sarcoma, other soft tissue sarcoma, neuroblastoma, and nephroblastoma PDX. Tumor volumes depicted were obtained from 2 to 14 animals per PDX model. Tx: transplantation; * - eRMS, ** - other RMS. Error bars represent ± standard error of mean (SEM).

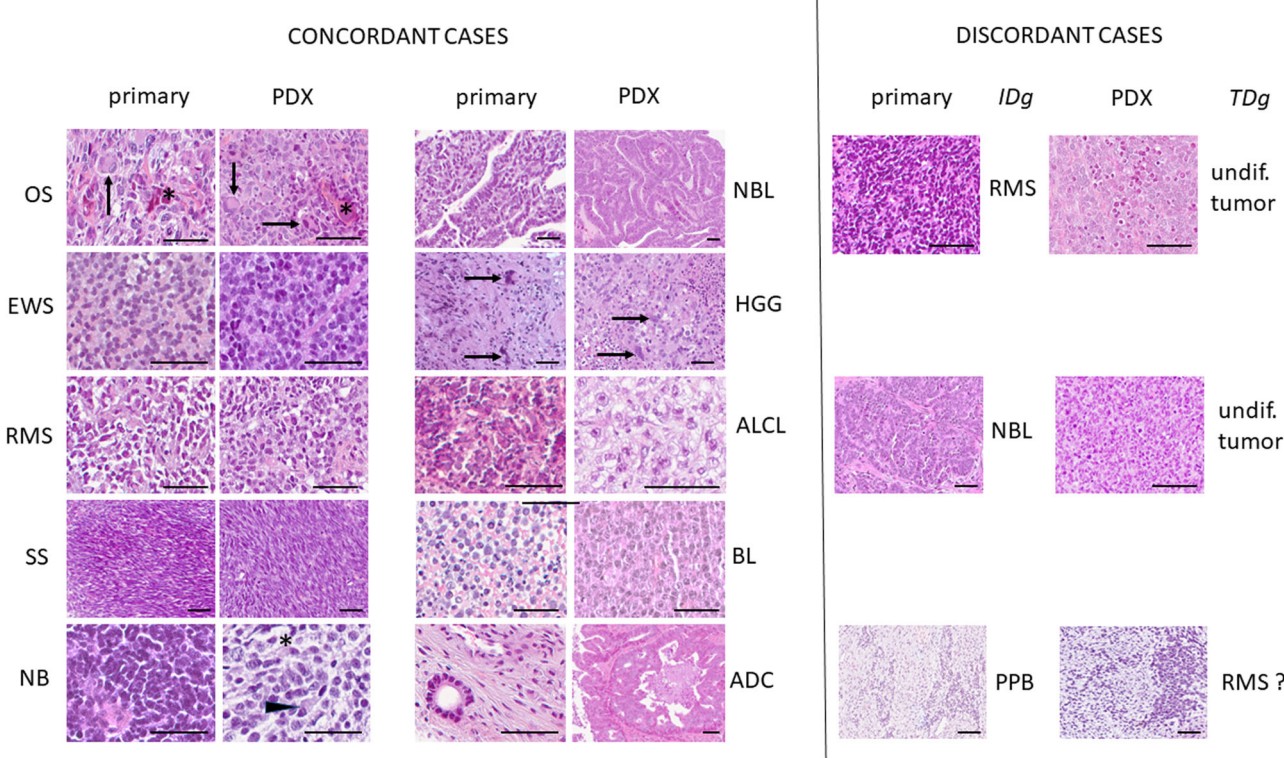

**Fig. 3 Comparative histological study between primary and PDX tumors.** Concordant cases: all PDX tumors have a suggestive morphological appearance, as seen in the case of synovial sarcoma (SS), identical to the primary, and that of neuroblastoma (NB), with neuropil (*) and the presence of large cells with eccentric nuclei (arrowhead). Note the striking similarities in some cases. In the case of osteosarcoma (OS), the PDX tumor retains the osteoid formation (*) and the presence of giant cells (arrows) observed in the primary; in the case of glioblastoma (HGG), the giant cells (arrows) observed in the primary are visible also in the PDX tumor. In the case of adenocarcinoma (ADC), only a bone biopsy was available for comparison: this likely explains the morphological differences observed between the primary and the PDX tumor, in which, however, the features are highly suggestive that the diagnosis of adenocarcinoma cannot be missed. Discordant cases: the examples illustrate the three main causes of discrepancies between the initial diagnosis (IDg) and the PDX-based tentative morphological diagnosis (TDg). In this case of rhabdomyosarcoma (RMS) suggestive histological features are absent and the PDX tumor looks morphologically undifferentiated (undif.). In this case of nephroblastoma (NBL), the PDX tumor is made only by "blastic" cells and lacks the distinctive epithelial structures present in the primary. In pleuropulmonary blastoma (PPB), the correct diagnosis cannot be achieved by morphology alone. Hematoxylin-eosin-saffron staining; scale bar = 200 μm. Additional abbreviations: EWS Ewing sarcoma, ALCL anaplastic lymphoma, BL Burkitt's lymphoma.

**Molecular characteristics in PDX and primary tumors**. To further evaluate the faithfulness of these models, we compared WES and RNA-Seq of the primary tumors with their derived PDX focusing here on 39 PDX models from Gustave Roussy. The genome/transcriptome aligned on human (tumor clones) and mouse (tumor microenvironment) sequences were analyzed separately using Xenome. Recurrent alterations, found in both the patient tumor sample and the corresponding PDX, and considered by the clinical molecular tumor board as actionable or of interest in the disease were identified (Fig. 4a). PDX models exhibited high consistency of the genetic alterations found in the patient samples. Main discordance in molecular phenotyping was observed in the WES of GR-RMS-1 where the *PIK3CA* and *TP53* pathogenic variants and the *MYCN* amplification were not detected, whereas the *PAX3::FOXO1* gene fusion was present in the RNA-Seq analysis suggesting a technical concern with the WES analysis. The GR-BALL-1 PDX with distinct pattern had been engrafted with a minor clone (see leukemia chapter). The Burkitt lymphoma GR-BL-1 shares the *TP53* mutation with the primary tumor but not the *MYC* variant.

Taking advantage of the PDX model, a human tumor growing in a mouse microenvironment, we deconvoluted bulk RNA-Seq of matched patient/PDX samples to isolate the transcriptomic program of tumor cells from the microenvironment in both patient and PDX tumors (see Methods). Biased by the sampling and the tumor purity, the quantification of the alteration conservation with the Jaccard distance between matched patient and PDX suggests that osteosarcoma and rhabdomyosarcoma are the most divergent, in accordance with their unstable genetic (Supplementary Fig. S3a and b). Transcriptional landscape of tumor cells, summarized by five Tumor Principal Components (TPCs; Supplementary Fig. S3c), is highlighted after functional enrichment analysis of the tumor cell origins for each pathology (Supplementary Fig. S3d). Clustering of patient and PDX samples in this tumor latent space, is emphasized by the UMAP of the Fig. 4b, suggesting for 97% of the patients a high conservation of tumor cell phenotypic traits from primary tumors to PDX. The single exceptions are non-Hodgkin lymphoma or non-rhabdomyosarcoma soft tissue sarcoma, both classified by this methodology near osteosarcoma. This result is probably explained by the small number of patients for these pathologies, a challenge for any attempt of phenotypic characterization with bulk RNA-Seq data. Deconvoluting the microenvironment present in both patient and PDX models, summarized by the Microenvironment Principal Components (MPCs, Supplementary Fig. S3e, S3f and S3g), did not demonstrate any clear-cut conservation of the microenvironment (Supplementary Fig. S3e) in mice despite an interesting trend in osteosarcoma to cluster

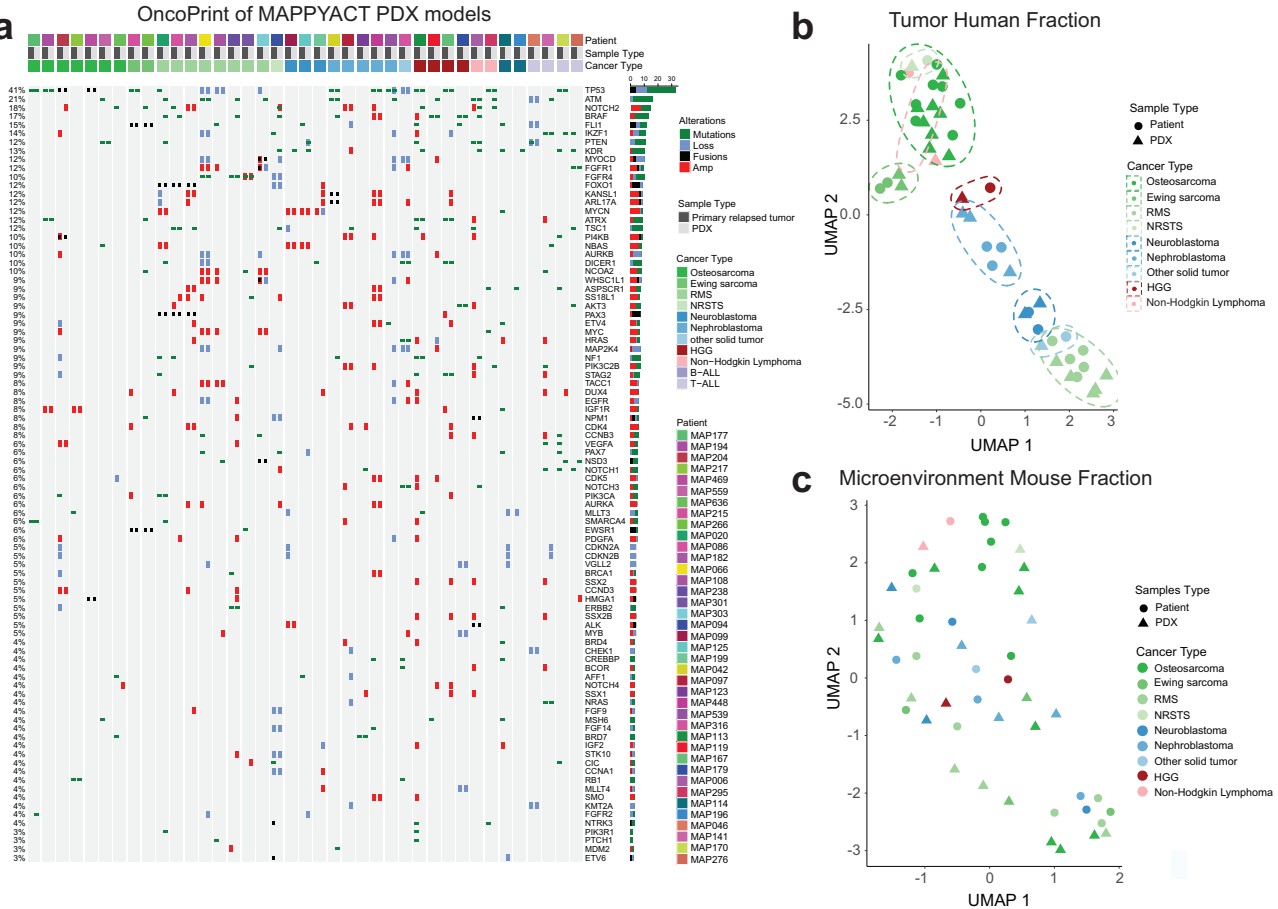

**Fig. 4 Pangenomic profile and comparison of WES and RNA-Seq between matched primary patient tumors and PDX model. a** OncoPrint presents tumor specific alterations and genes considered as actionable or of interest in 39 primary tumor sample and PDX, respectively. **b** Expression of human genes cluster PDX together with primary tumor histology types. **c** Microenvironment of murine species in patient and PDX samples that is further specified in Supplementary Fig. S3.

together suggesting a partial reconstitution of osteosarcoma microenvironment in PDX. This clustering of the osteosarcoma microenvironment fraction in mice and patients is driven by the MPC1 (Supplementary Fig. S3f), and is identified as the regulation of the immune response by functional enrichment analysis (Supplementary Fig. S3h). Most contributing genes to the MPC1 such as *IL10RA*, *CD48*, *SLA*, suggest that MPC1 describes the antigen presenting cell (APC) role in the immunosuppressive prone context of osteosarcoma relapse (MPC1 wordcloud in Supplementary Fig. S3i).

**HLA class I typing and allele and supertype coverage of the established PDX models.** In the emerging era of personalized cancer immunotherapy, the characterization of HLA alleles is a critical aspect for preclinical studies using PDX models such as adoptive transfer of neoantigen-specific T cells. Recently, we characterized the HLA class I and II genotypes of patients from the MOSCATO-01[19] and MAPPYACTS trials[2]. Thus we first compared HLA class I (HLA-A, -B, -C) genotypes for the 34 PDX models of solid tumors developed at Gustave Roussy with available WES and RNA-Seq data with their equivalent patient normal and primary tumor samples, using a combination of dedicated HLA typing algorithms (see Methods). Because of their consistency, we inferred the HLA genotypes for all solid tumor PDX models developed in MAPPYACTS from the corresponding patient sequencing data. Overall, 27, 47 and 28 HLA-A, -B, and -C alleles, respectively, were detected among the 110 PDX models established

from patients with solid tumors and available normal WES data (Fig. 5, Supplementary Fig. S4 and Supplementary Data 2). When considering PDX models with available RNA-Seq in comparison to normal WES, all alleles from the three classical HLA class I loci were detected in Ewing sarcoma, non-rhabdomyosarcoma soft tissue sarcoma and nephroblastoma PDXs, and all except single HLA-B and -C alleles were detected in alveolar rhabdomyosarcoma and high-grade glioma PDXs. In contrast, only 2 out of 6 HLA-A alleles and none of the HLA-B and-C alleles were detected from RNA-Seq of neuroblastoma PDXs, consistent with low HLA-I transcriptional levels in this tumor type[20,21] (Supplementary Fig. S4 and Supplementary Data 2).

Furthermore, we looked at the genetic ancestry fractions of all patients included in the trials using the EthSEQ pipeline and compared the distribution of HLA allele frequencies in the PDX models with those observed in our European (EUR) patients study population. The cumulative HLA allele frequency coverage per tumor type or subtype in the corresponding PDX models ranged from 0.40 to 0.96 for HLA-A, from 0.27 to 0.80 for HLA-B, and from 0.23 to 0.97 for HLA-C (Supplementary Fig. S4). Notably, the HLA allele frequency coverage was >0.6 for both HLA-A and -C genes and ranged from 0.4 to 0.80 for HLA-B in the PDX panels representing the main pediatric solid tumor types and subtypes, with the exception of other rhabdomyosarcoma, carcinoma, hepatoblastoma, pleuropulmonary blastoma, ependymoma and atypical teratoid rhabdoid tumor, with limited numbers of models (*n* = 1–3).

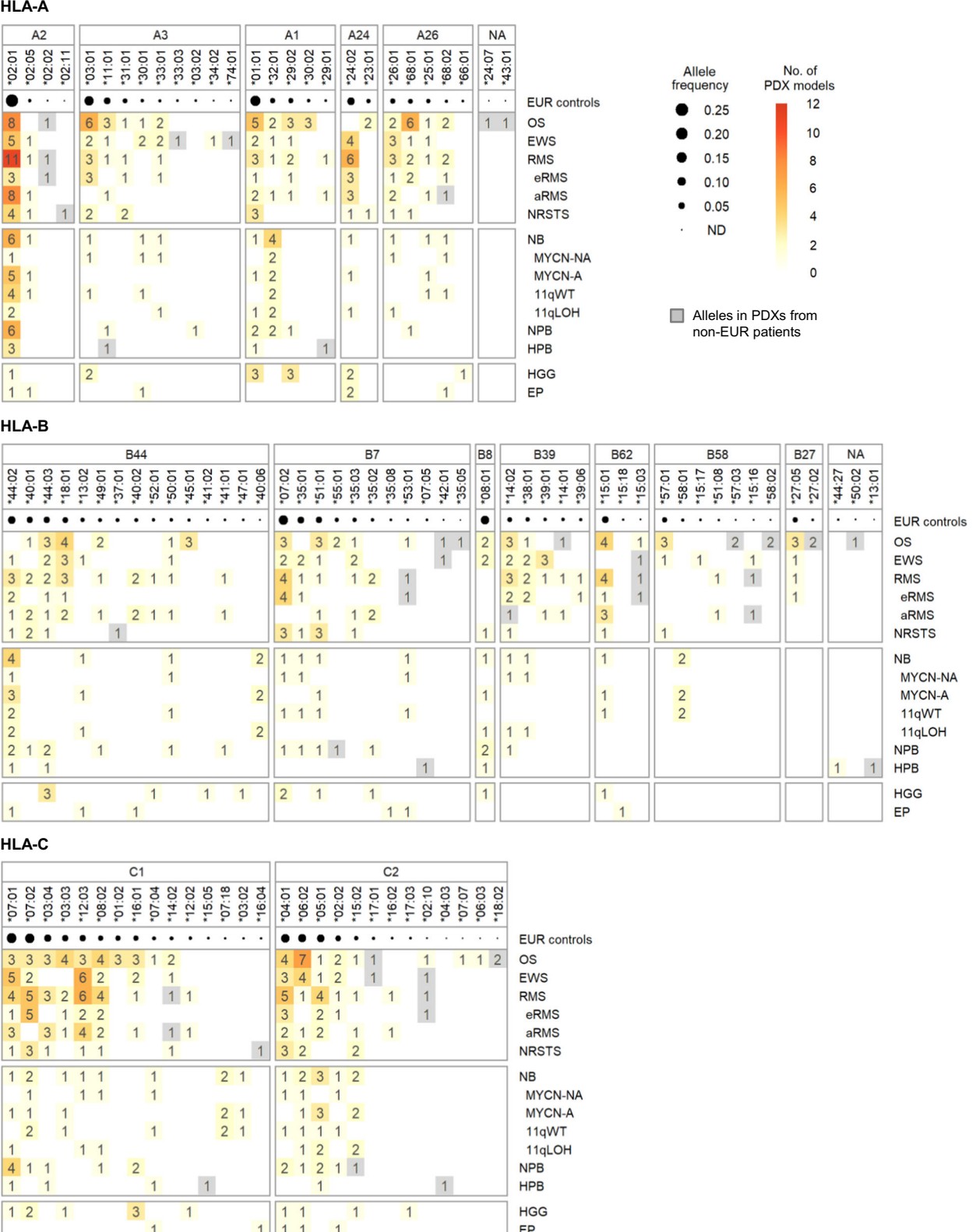

**Fig. 5 HLA class I alleles detected in PDX models from pediatric solid cancers.** For each tumor type and molecular subtype, the numbers of alleles in PDX models harboring the corresponding HLA-A, -B and -C alleles are indicated within the boxes (1–11). HLA alleles were grouped according to their supertype-specific binding motifs, unless unclassified (denoted with "NA"), which are indicated at each HLA gene panel, and sorted by descending order of allele frequencies reported from EUR control individuals. Alleles detected only in PDXs derived from non-European (non-EUR) individuals are shown in gray boxes.

It has been shown that the peptide-binding specificities of multiple HLA class I alleles largely overlap and can be grouped as supertype-specific binding motifs, corresponding to 5 HLA-A (A1, A2, A3, A24, A26), 7 HLA-B (B7, B8, B27, B39, B44, B58, B62) and two HLA-C (C1, C2) supertypes[22,23]. A total of 97 out of 102 (95.1%) HLA-I alleles detected in the PDX models could be assigned to known supertypes, with only three alleles from osteosarcoma (A*24:07, A*43:01 and B*50:02) and two from hepatoblastoma (B*13:01 and B*44:27) unclassified. HLA-I supertypes were largely represented among the PDX models (Fig. 5, Supplementary Fig. S5 and Supplementary Table S3): all 5 HLA-A supertypes in sarcoma, neuroblastoma, high-grade glioma PDXs, and 5–7 HLA-B supertypes in sarcoma and neuroblastoma. Both HLA-C1 and -C2 supertypes were represented in the PDXs corresponding to all specific tumor types and subtypes.

**Metabolic signatures in PDX maintain consistency with primary tumors.** Metabolic reprogramming has long been identified as a hallmark in cancer[24]. This dynamic reprogramming is driven by tumor intrinsic and extrinsic factors, such as microenvironment and nutrient availability. We investigated the conservation of metabolic networks by evaluating the metabolic capacity of the human fraction of the PDX and primary tumors to carry defined metabolic tasks (Fig. 6a). We inferred metabolic function (=performing metabolic tasks) by quantitatively evaluating the expression of essential enzymes to carry a set of core metabolic tasks. Comparison of metabolic network content showed similar metabolic function of PDX to their corresponding cancer types (Fig. 6b). Highly significant similarity ($p = 5.48e−10$) was found comparing matched patient primary tumor-PDX to randomly selected pairs (Fig. 6b). After confirming maintenance of global metabolic reprogramming phenotype, we investigated whether specific metabolic tasks are conserved. Figure 6c shows an example, the synthesis of dopamine from tyrosine that is present in the two neuroblastoma patient-PDX pairs. Capacity to fulfill this metabolic task can be expected in those tumors originating from adrenal gland. Interestingly, one carcinoma pair also showed the capacity to make dopamine. We expanded this analysis to all PDXs, including the ones without matched primary tumor, and found higher task score only in neuroblastoma and few other cancers, including HGG that can be expected to be derived from dopaminergic neurons (Fig. 6d). This confirms the ability of the PDX model system to reproduce expected metabolic functions and describe unexpected findings. We next evaluated the capacity of tumors to synthesize L-kynurenine from tryptophan, an oncometabolite known to exert immunosuppressive effects[25]. We found that L-kynurenine can be produced by many cancer types, most strikingly in sarcomas, including osteosarcoma. Its synthesis was confirmed by numerous corresponding PDXs, which can be used in follow-up work to investigate its immunosuppression role and potential therapeutic implications (Fig. 6e).

**Selected tumor types of PDX models.** A description of osteosarcoma PDX developed at Gustave Roussy, both as subcutaneous and as orthotopic (paratibial) models, and their comparison with the originated patient sample at relapse and the primary tumor at diagnosis are presented in a separate manuscript[26]. We further wished to provide additional context on models from selected pediatric tumor types for which multiple PDX models were established or for some tumor types that have so far rarely been reported.

**Acute Leukemia PDX.** Leukemia is the most common type of cancer in children and, despite significant advances in treatment, it remains the second cause of cancer-related deaths. In this study, we transplanted 27 'relapsed' acute leukemia samples into sublethal irradiated NSG mice and established 15 models of childhood acute leukemias (7 B-ALL, 6 T-ALL, 2 AML-M1 and -M4/5; 56% successful engraftment rate overall with special difficulty for AML (20% versus 78% and 75% for B- and T-ALL respectively). Phenotypic analyses of the B-ALL models are depicted in Supplementary Fig. S6 and show that the major phenotypic clone, based on CD34/CD38 expression, was amplified in NSG recipients established from GR-BALL-2, GR-BALL-3 and GR-BALL-7 samples. Two B-ALL models were engrafted with minor clones: GR-BALL-1 and GR-BALL-8; the GR-BALL-1 profile indeed looks distinct from the originating primary leukemia profile (Fig. 4a). Notably, we also established a B-ALL cell line from the GR-BALL-3 sample, validated at the phenotypic and genomic levels (Supplementary Fig. S6b and c), in which expression of the TCF3::HLF fusion transcript associated with aggressive leukemia has been confirmed (Supplementary Fig. S6d). Together, although a relatively low number of leukemia samples where included in this study, however we developed models of aggressive leukemia.

**Rhabdomyosarcoma (RMS).** RMS is the most common type of soft-tissue sarcoma of childhood with a poor prognosis in patients with metastatic or recurrent disease[27]. The most frequent subtypes are embryonal (eRMS) and alveolar RMS (aRMS) in ~70% and 30% of childhood rhabdomyosarcoma, respectively. aRMS are characterized by FOXO1 gene fusions resulting from the stable reciprocal translocation of chromosomes 1 or 2 and 13 encoding two paired box transcription factors PAX7 and PAX3, respectively, while RMS that are FOXO1 fusion-negative harbor mutations in key signaling pathways[28].

Here we established 25 RMS PDX models, 12 FOXO1 fusion-positive aRMS, 9 eRMS and one each epithelioid, sclerosing, spindle cell and NOS. All had previously been treated with chemotherapy; 14 were from initially well responding tumors that relapsed, 11 were from primary refractory diseases. Ten models were derived from primary recurrent/refractory tumor sites, 15 from metastases. Ten of 12 established PDX exhibited morphological features concordant with their derived patient tumor (Supplementary Table S2). All aRMS PDX maintained the PAX3::FOXO1 fusion; aggressive features like TP53 mutations ± loss of heterozygosity, FGFR4 mutations ± amplification, PIK3CA or NRAS mutations, and cell cycle alterations like CDK4 amplification or CDK2NA/B loss were present in the rhabdomyosarcoma PDX. The sclerosing RMS PDX exhibits NSD3::FGFR1 and NSD3 (WHSC1L1)::MYOCD fusions and the spindle cell RMS the characteristic VGLL::NCOA2 fusion.

**Ewing sarcoma.** Ewing sarcoma is the second most frequent bone tumor of childhood and adolescence after osteosarcoma. Genetically it is characterized by a balanced chromosomal translocation involving EWSR1 or FEV and a gene of the ETS transcription factor family, with the most common fusion being EWSR1::FLI1. Other recurrent genetic alterations are infrequent except for STAG2 and TP53 mutations and CDKN2A deletions[29]. Combination of these three alterations with the translocation appear to favor the oncogenic transformation of Ewing sarcoma cell of origin[30]. Patients with STAG2 mutations alone or associated with TP53 mutations have a lower overall survival as compared to those with a wild type status[29]. Mechanistically, STAG2 loss of function favors invasiveness and metastasis of Ewing sarcoma cells by reducing cis-mediated EWSR1-FLI1 activity[31]. Fifteen Ewing sarcoma PDX were established, 13

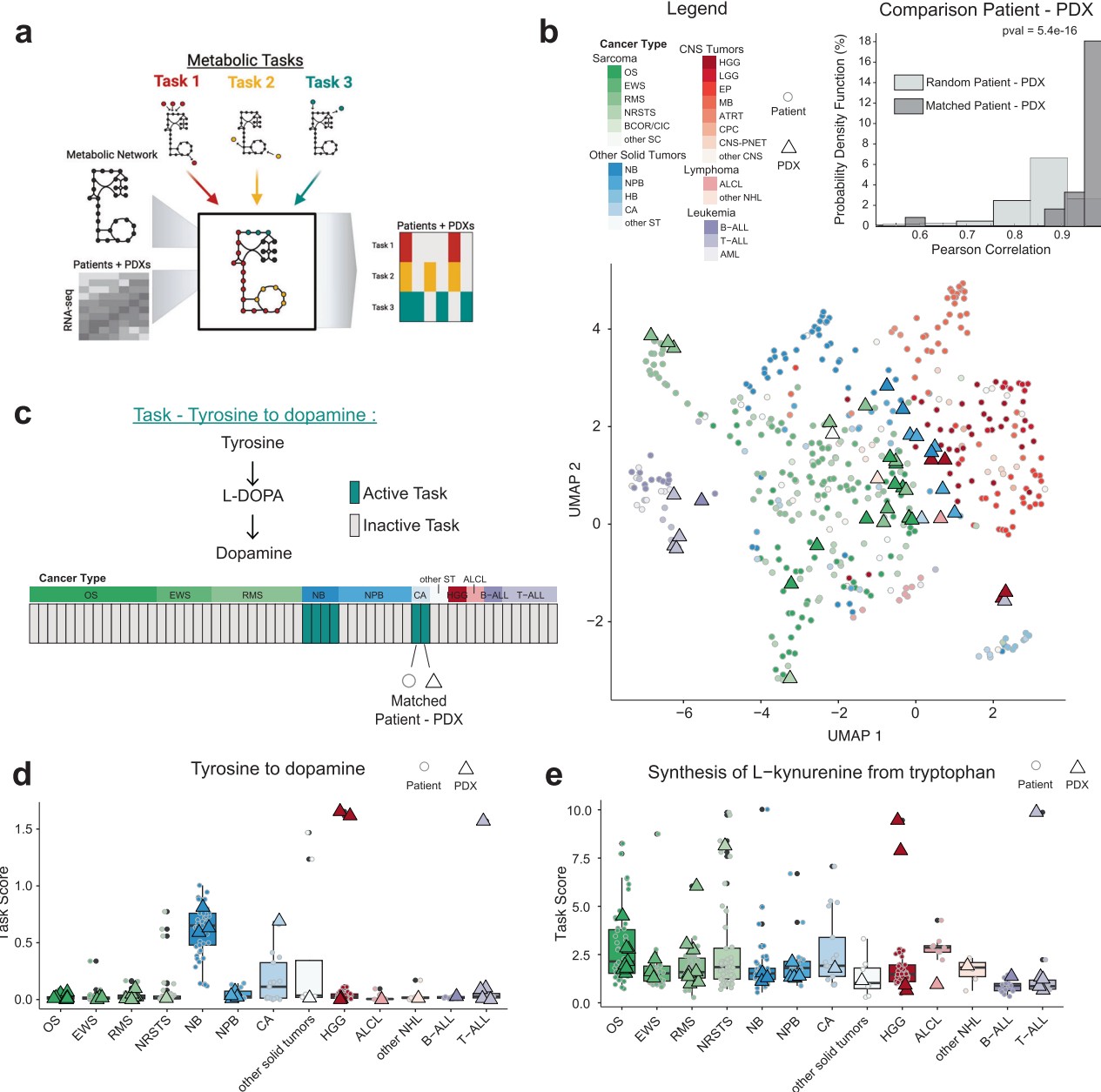

**Fig. 6 Metabolic functional signatures of 42 PDX models depicted on the background of the whole MAPPYACTS patient population. a** Metabolic functional analysis of PDX and patient tumors using Cellfie, which evaluates the capacity of cancers to carry out 195 metabolic tasks. **b** Global comparison of PDXs and patients metabolic function using UMAP and correlation distribution between matched and random patient-PDX pairs (Student *t* test). **c** Tyrosine to dopamine metabolic task and its binary score. Displayed are matched patient-PDX pairs ordered and colored in green if task is active. **d** Tyrosine to dopamine task score for all cancer types in patients (circles) with PDX models (triangles). **e** Synthesis of L-kynurenine from tryptophan task score of patients and PDX models. B-ALL/T-ALL acute lymphoblastic leukemia, AML acute myeloid leukemia, ALCL anaplastic large cell lymphoma, ATRT Atypical teratoid rhabdoid tumor, BCOR/CIC BCOR or CIC-translocated sarcoma, CA carcinoma, CNS central nervous system tumor, CPC choroid plexus carcinoma, EP ependymoma, EWS Ewing sarcoma, HB hepatoblastoma, HGG high-grade glioma, LGG low grade glioma, MB medulloblastoma, NBL neuroblastoma, NPB nephroblastoma, NHL non-Hodgkin lymphoma, NRSTS non-rhabdomyosarcoma soft tissue sarcoma, OS osteosarcoma, PNET primary neuroectodermal tumor, RMS rhabdomyosarcoma, RT rhabdoid tumor, SC sarcoma, ST solid tumor.

displayed the *EWSR1::FLI1* translocation, one each *FUS::ERG* and *EWSR1::FEV*. PDX models display *TP53* mutation/LOH, *STAG2* mutation or deletion or *CDKN2A* deletion; all exhibit the typical small round cell morphology.

**Nephroblastoma**. Nephroblastoma or Wilms tumor is the most frequent renal malignancy in childhood and occurs mainly in young children[32]. Nephroblastoma is considered a failure of embryonic kidney development, a complex process in which

different transcription factors, proto-oncogenes, and various types of growth factors are effective[32]. A recent initiative develops organoids from primary nephroblastoma within the UMBRELLA registry[33], however there is a scarcity of preclinical models overall and particularly at recurrence. In our hands, nephroblastoma had one of the highest rates for PDX establishment (73%) where 8/11 transplanted samples engrafted in P0 and were stable. Established PDX exhibited a variable morphological pattern; in the four cases examined histologically, two retained the distinctive epithelial

structures typical of nephroblastoma whereas two exhibited only undifferentiated, "blastic" tumor cell morphology.

**Pleuropulmonary blastoma (PPB).** PPB is a very rare, highly aggressive and malignant tumor, with poor prognosis originating from either the lungs or pleura. It occurs mainly in children of <5 or 6-years, but may rarely occur in adults[34]. From four patients with PPB, three samples were transplanted, two engrafted and were stable. Both established models are from 4-year-old patients with *DICER1* germline mutations with tumors displaying an additional somatic *DICER1* pathogenic variant; both also showed *NOTCH3* pathogenic mutations, one of them combined with a *KRAS* and *TP53* hot spot mutation, or a *HRAS* mutation with *BCOR* and *KAT6A* mutations, respectively.

**High-grade glioma (HGG).** Gliomas are the most common primary tumors of the brain and account for almost half of all CNS tumors in children and adolescents. HGG are aggressive, malignant lesions, accounting for 8–12% of all pediatric CNS tumors and of highly heterogenous nature distinct from adult HGG. Molecular characteristics mainly distinguish tumors harboring histone mutations, *H3.K27M* in midline and *H3.G34* in hemispheric, tumors harboring the *BRAF* V600E mutation, and *H3-/IDH* wild type tumors[35]. The prognosis for patients with pediatric HGG remains poor with limited treatment options.

Three of the six established HGG are pleomorphic xanthoastrocytoma, all with anaplastic features. Two of them exhibited the characteristic *BRAF* p.Val600Glu mutation and had homozygotic focal deletions of *CDKN2A/2B*. The third exhibited a *TP53* p.Arg337Cys hot spot mutation as well as a pathogenic variant in *TSC2*, *ATRX* and *PTPN20A*. In contrast, all three other HGG exhibited alterations in the PI3K pathway. The relapsed primary hemispheric giant cell glioblastoma had a *PIK3R1* pathogenic variant, associated with pathogenic variants of *NF1*, *ATRX*, and a heterozygotic *TP53* mutation. The second relapsed temporal glioblastoma displayed a *PIK3CA* mutation, a *TSC1* VUS, focal homozygotic deletion of *NF1* and *CDKN2A/B*, as well as *mTOR* germline VUS associated with an isodisomy. The pontine glioma exhibited a *PIK3CA* hot spot mutation and pathogenic variants in *CDKN2C* and *ACVR1*. However, despite a loss of trimethylation of lysine 27 of histone H3 in immunohistochemistry, the classically associated *H3.1* mutation was not detected. These three could be excellent models to further explore the role of targeting the *PI3K/AKT/mTOR* pathway. Specific inhibitors at all three levels are in clinical development and the optimal agent to be taken forward has not yet been defined, as we recently discussed in the paper exploring a dual *mTORC1/2* inhibitor in AcSé-ESMART[36]. The *BRAF* mutated anaplastic xanto-astrocytoma will contribute to the understanding of resistance mechanisms to *BRAF ± MEK* targeting that is not yet elucidated in pediatric glioma[37,38].

## Discussion

Here we report on successful large-scale integration of a pediatric PDX development program within an advanced clinical molecular profiling trial (MAPPYACTS)[2]. This academic initiative established to date 131 unique PDX models, characterized by deep phenotyping, all derived from recurrent or refractory pediatric cancers. This focus on advanced and high-risk pediatric cancers, provides high quality models at large scale to help tackle the major challenges in pediatric oncology. The ultimate aim of this academic initiative is to foster use of these models for basic and translational research. We therefore provide an extensive characterization of the primary patient's tumors and matched PDX. Importantly, 90 of the 131 models presented here have been

shared with the IMI2 ITCC-P4. This European Union-supported project aims at performing preclinical drug development at high-scale through an academic-industry partnership following additional molecular characterizations. With this combination, we are confident to lay the basis for rich academic and industrial follow up projects and foster translational development.

Pediatric cancers with single unique key driver events are rare[2,39] and the challenge of pediatric precision medicine programs is addressing the underlying cancer complexity with innovative therapeutic approaches. Likely combination strategies targeting several hallmarks of the cancer at the same time will open new avenues. As the established PDX models retain tumor heterogeneity they allow to explore co-targeting strategies with innovative therapies in a preclinical setting before treating children. We further extended the characterization of our models to two therapeutic domains that have been insufficiently explored in pediatric cancers. First, the characterization of all models for HLA status, which has been neglected so far in PDX reports, and which allows to select the appropriate model for immune therapy development. Our PDX capture most of the HLA class I allelic and supertype diversity observed in cohorts of patients with pediatric solid tumors, providing a panel of relevant preclinical models to evaluate HLA-restricted T-cell based therapeutics in vivo. Furthermore, the models harboring potential loss or downregulation of HLA expression such as neuroblastoma PDXs could be useful to assess strategies to restore tumor HLA expression and improve T cell recognition. Second, to prepare preclinical evaluations of metabolic inhibitor candidates we included a preliminary characterization of our models in regards to metabolic signatures. To our knowledge, these aspects have not been addressed in other PDX programs.

This pediatric PDX study highlights several essential aspects critical for its success. First, from a clinical point of view, it is important to highlight that nearly all patients and parents supported the development of a preclinical model. The ethical aspects of consenting to preclinical models were extensively discussed by Smith and colleagues[40]. In our clinical trial, the primary aim of MAPPYACTS was to perform a characterization of the recurrent/refractory tumor to provide a molecularly guided treatment suggestion to the patient. The development of PDX was an ancillary project that was based on supplementary tumor samples without an additional intervention. This fact, as well as the advanced disease and the high medical need of patients may have contributed to the high acceptance rate of this project.

Second, it highlights the importance of a functional network of academic centers that are experienced in pediatric PDX establishment. Close collaboration between the research laboratories and clinical partners was essential for the high rate of successful PDX establishment. All research laboratories contributed with their own resources to take this opportunity forward and thanks to multiple external funding, mostly charities. Material and data transfer agreements have been set up through institutional IP Transfer units to allow sharing of the models with academic institutions.

Third, we identified the following main hurdles for successful xenotransplantation: small size of tumor biopsies in certain cases and the logistics in providing fresh tumor materials to the research laboratories. This was especially critical during the initiation of the project while training partners in standardized procedures and when interventions were needed to be performed rapidly due to a medical emergency. Despite the high enthusiasm of all partners, PDX model development is highly time and cost consuming, still limiting its integration in standard workflows. We approached these challenges by providing optional soft-

freezing boxes to the centers that could be used when primary tumor samples were not shipped immediately. This enabled to plan transplantations according to favorable time schedules and prioritization of high priority tumors. Confronted with limited resources, 223 samples have been soft-frozen in the laboratories and are still available for further model development based on primary cancer characteristics upon request.

Several of our models already have been used for preclinical explorations and were included in publications demonstrating the proof-of-concept for their relevance. Among those were Burkitt lymphoma[41], anaplastic large cell lymphoma[42], neuroblastoma[43–45], Ewing sarcoma[46], rhabdomyosarcoma[47,48] and acute lymphatic leukemia[49] PDX models.

We have generated a PDX tumor bank of 131 advanced pediatric tumors to implement at large scale modeling of pediatric relapsed and refractory tumors. These preclinical models not only provide a renewable source of biological material, but also allow to evaluate new anticancer drugs, therapeutic combinations and identify biomarkers in the most relevant pediatric cancer cohort (refractory/relapsed). This unique program sets new standards to develop PDX models within a pediatric precision medicine clinical trial. Our panel not only includes PDX models of various high-risk tumors but provides a well-annotated panel of matched PDX-clinical datasets. It therefore provides a resource for the cancer community to study basic biology questions and evaluate treatment strategies at large scale.

## Materials and methods

**Pediatric human tumor sample collection**. Patients with recurrent or refractory pediatric cancer underwent following informed consent a tumor biopsy, surgical resection, blood or bone marrow sampling for molecular characterization within the MAPPYACTS trial[2] in 18 medical centers in France, Ireland, Italy and Spain. Main inclusion criteria were age below 18 years at diagnosis, refractory or relapse, evaluable or measure disease at inclusion, good clinical performance status and life expectancy more than 3 months, no organ toxicity more than grade 1 and potential eligible to an early clinical trial. The development of preclinical tumor models was optional and performed only for the patients with specific consent and only if sufficient material was available. At intervention, additional tumor samples were collected in sterile falcon tubes containing DMEM medium with 1% penicillin-streptomycin or in single-use 100 mL RPMI 1640 medium bottle for solid tumors or empty sterile falcon tubes for bone marrow and blood samples. Samples were sent at room temperature within 48 h or soft-frozen in fetal bovine serum (FBS) with 10% DMSO and sent on dry ice to the corresponding research laboratory for PDX development. For bone marrow and blood samples a Ficoll separation in Phosphate Buffered Saline or Hank's Balanced Salt solution was performed after reception. The PDX study was non randomized, perspective, open-labeled and used a descriptive design.

**Experimental in vivo PDX development**. Animal care and use were performed in accordance with international guidelines and the recommendations of the European Community (2010/63/UE). Experimental procedures were specifically approved by the ethics committee, the France Ministry of Agriculture or Italian Ministry of Health; Gustave Roussy CEEA26 (CEEA PdL N°6, approval number: 2015032614359689 V7, 1281.01, C75-05-18, 2012-017), Institut Curie CEEA-IC #118 (APAFIS#11206-2017090816044613-v2), Centre Léon Bérard CEEA CECCAPP N°15, (APAFIS#10079), CEA (APAFIS#9458-2017033110277117 v2), Nantes University (APAFIS#32043-2021061811307790 v2), University Strasbourg (APAFIS #2017021410378167),

Fondazione IRCCS Istituto Nazionale dei Tumori (OPBA authorization: INT 03_2018, Italian Ministry of Health authorization: 646/2018-PR), IDIBELL animal facility committee (AAALAC Unit1155).

For solid tumors, heterotopic and/or orthotopic PDX were established depending on the tumor size and tumor histology in 3–7 weeks immunocompromised female and male Swiss athymic Nude (Crl:NU(Ico)-Foxn1nu), SCID (CB17/Icr-Prkdcscid/IcrIcoCrl), NSG (NOD.Cg-PrkdcscidIL2rgtm1Wjl/SzJ) or NSG expressing human cytokines (NSG-IL) mice, obtained from the institutional animal facilities or from Charles River animal facilities, as reported previously[50–53]. Tumor samples of 2–5 $mm^3$ were implanted subcutaneously (SC) on one or both flanks, in the sub interscapular fat pad (FP), paratibially with periosteum activation (PT)[53] (for bone tumors), intramuscular (IM) (rhabdomyosarcoma), or in the left kidney capsule (KC) (samples smaller than 2 $mm^3$) into 2–5 mice. For central nervous system (CNS) tumors, tumor samples were digested and cell homogenates injected intracerebral into the caudate nucleate, the cerebellum or the pons (IC) of nude mice. For leukemia samples, mononuclear cells (50,000 to $10^6$ cells) were either directly injected into the femur or intravenously transplanted in sub-lethally irradiated (2.5 Gy) NSG mice. Following injection of patients' cells into primary recipients, bone marrow and blood samplings were performed and the human leukemic cell infiltration was measured by flow cytometry using the standard antibodies for the following markers (T-ALL: hCD45, hCD7; B-ALL: hCD45, hCD34, hCD38, hCD19 and hCD10; AML: hCD45, hCD34, hKIT, hCD33). For T-ALL: When %hCD45 + CD7+ leukemic cells >80%: mice were euthanatized and cells recovered from mouse femurs. For B-ALL and AML: endpoint criteria include a primary engraftment within <6 months and a percentage of blasts positive for one of the indicated markers >20% in primary recipients. 1–3 × $10^6$ cells from the bone marrow of primary recipients were then transplanted into secondary (P2) recipients. For P2, the % of blasts in the bone marrow was retained as the main criteria (>80% at endpoint). For each model, the injection route and mouse type are indicated in Supplementary Data 1.

When the tumor successfully engrafted in mice, in vivo passages (P) were performed in order to amplify and stabilize the PDX growth (≥ P2). For acute T-cell leukemia, the model was considered established after the first growth in mice. A thawing test from P2/P3 soft-frozen PDX was performed to certify the master stock.

Clinical status, tumor take and growth were evaluated one to three times a week. Subcutaneous, fat pad and paratibial xenografts were detected by palpation and measured by calipers; kidney capsule engraftment was confirmed with an Aplio XG ultrasound equipped with a probe of high frequency wide band (7–14 MHz; LTP 1202; Toshiba), performed under anesthesia with 3% isoflurane, and brain tumors by clinical symptoms observation. Subcutaneous xenograft volume was calculated according to the equation: V ($mm^3$) = width$^2$ ($mm^2$) × length (mm)/2. The experiments lasted until tumors reached specific endpoints detailed in the ethical projects. Tumor doubling time (Td) was determined in an exponential growth phase between 200 and 400 $mm^3$. For brain tumors, xenograft take was based on appearance of clinical symptoms and body weight loss (~10% in 24 h) which determined the endpoint and survival curves were established. For leukemia, blast engraftment was detected by blood count analysis. PDX were morphologically (histology) and molecularly characterized (RNAseq and WES) and compared to the patient's tumor.

**PDX characterization**. PDX were cryopreserved for banking, morphologically (histology) and molecularly characterized

(RNAseq and WES) and compared to the patient's tumor, as detailed in the chapters below. Scripts are available on demand and can be upload on a public GitHub server.

Selected solid tumor PDX models from Gustave Roussy ($n = 44$), Institut Curie ($n = 41$), XenTech ($n = 3$) and Istituto Nazionale dei Tumori Milan ($n = 2$) were shared with the European Innovative Medicines Initiative 2 (IMI2) ITCC-P4 (Innovative Therapies for Children with Cancer-Pediatric Preclinical Proof-of-concept Platform) initiative (Project Number: 116064; IMI2/INT/2015-03842 v.2020; https://www.itccp4.eu/), in which these centers participate. The models were established based on the MAPPYACTS program and consents, and will contribute also to the European IMI2 ITCC-P4 program.

**Statistics and reproducibility.** The design of the study was descriptive with basic standard calculations, using Graphpad Prism® Software version 9.00 (Graphpad Software Inc, La Jolia, CA, USA) and R for graphical presentations.

**Tumor cryopreservation.** Tumor fragments that were not utilized for in vivo passaging, RNA/DNA extraction or histology were cryopreserved for banking and later usage. Tumor fragments (at least 3 in each tube) and leukemic blasts (at least 2 tubes/patient with 10–25 million) were placed in 1 mL of FBS/10% DMSO per tube, soft-frozen at −80 °C and transferred to liquid nitrogen for long-term storage.

**Histology and immunohistochemistry.** Tumors were fixed in 4% paraformaldehyde and embedded in paraffin. Four μm sections were stained with hematoxylin-eosin-safranin (HES) for morphology and processed after heat-induced antigen retrieval procedure using a mouse anti-human Ki67h antibody (1:20; Dako Ref:M075501 clone MIB-1), mouse anti-human CD3 (1:20; Dako Ref:M7254 clone F7.2.38), CD20 antibodies (1:200; Dako Ref:M075501-2 clone L26), and relevant tumor type specific antibodies, visualized by Klear Mouse DAB kit (GBI Labs) and a Zeiss Axiophot microscope. Single representative whole tumor tissue section from each animal was digitized using a slide scanner NanoZoomer 2.0-HT (C9600-13, Hamamatsu Photonics) and reviewed by one pathologist (JYS).

**Molecular characterization.** Tumor cellularity was assessed on the primary tumor samples, tumor DNA and RNA and germline DNA were isolated and whole exome (WES) and RNA sequencing (RNA-Seq) performed within the main trial MAPPYACTS[2]. For the established PDX samples, whole exome was captured from 500 ng of PDX sample DNA using the Agilent SureSelect V5 (50 Mb) or Clinical Research Exome (54 Mb) kit. RNA sequencing libraries were prepared with TruSeq Stranded mRNA kit following recommendations. The key steps consist of PolyA mRNA capture with oligo dT beads 1 μg total RNA, fragmentation to ~400 pb, DNA double strand synthesis, and ligation of Illumina adaptors amplification of the library by PCR for sequencing. Sequencing of subsequent libraries was performed using Illumina sequencers (NextSeq 500 or Hiseq 2000/2500/4000) in 75 bp paired-end mode. Data were processed by bioinformatics analyses.

**Molecular abnormalities reporting.** WES and RNA-Seq results of the PDX samples were analyzed and compared to the patient's tumor analysis. In order to identify mouse reads from the PDX, reads from WES and RNAseq were first classified as of human (hg19) or mouse (mm10) origin with Xenome (v1.0). All WES reads classified as human were then aligned using BWA (0.7.12; BWA, RRID:SCR_010910). The variant calling was performed using Varscan2 (2.3.9; VARSCAN, RRID:SCR_006849). The copy number calling was performed using Sequenza (2.1.2). The variants are annotated with Annovar. Somatic alterations with <5 minimum reads supporting the mutations, 5% of the reads covering the sequence supporting the alteration and more than 1% of the population annotated with the mutation in the databases 1000g2015aug (latest 1000 Genomes Project dataset with allele frequencies in six populations including ALL, African, Admixed American, East Asian, European and South Asian) and kaviar_20150923 (latest Kaviar database with 170 million variants from 13 K genomes and 64 K exomes) were filtered out. Samples similarity estimation based on the somatic mutations was performed using Jaccard distance.

Quantification of gene expression from the RNA-Seq human fraction was estimated using Salmon (0.9.0) on the GENCODE reference transcriptome (v27). Gene fusion calling was performed using the nf-core rnafusion pipeline running Arriba, Star-Fusion, EricScript and Squid. The differential gene expression analysis was performed using the R package Deseq2 (DESeq2, RRID:SCR_015687).

**HLA class I typing.** Four-digit typing of classical HLA class I (HLA-A, -B, -C) alleles was performed for 34 PDX models of pediatric solid tumors established at Gustave Roussy with available WES and RNA-Seq, and from the corresponding NGS data of patient normal (WES) and primary tumor samples (PTS) (WES, RNA-Seq). For the other solid tumor models without currently available NGS data ($n = 76$), HLA class I genotypes were inferred from patient normal and PTS samples. Briefly, HLA typing was performed using HLA-HD[54] from WES and RNA-Seq, and using both HLA-HD and HLAProfiler[55] from RNA-Seq. When both algorithm identified a discordant genotype, HLA typing was repeated using Optitype, xHLA and HISAT-genotype[56–58], from patient normal and PDX WES. A final consensus genotype was deduced when two algorithms identified the same allele(s) from two different NGS samples. HLA allele frequencies were calculated for patients with a European (EUR) ancestry fraction of at least 70%, determined using the EthSEQ pipeline[59], as perfomed for all patients of the MOSCATO and MAPPYACTS trials. HLA-I alleles were assigned to known supertypes based on the corresponding binding motif specificities[22,23,60–62].

**Metabolic functional signatures.** Metabolic function analysis was performed using Cellfie algorithm[63] on the RNA-Seq of primary tumors and of the human fraction of the PDXs. Analysis was produced using Recon version 2.2 as reference metabolic model network. Parameter used for gene expression thresholding was 'minmaxmean', where the threshold for a gene is determined by the mean of expression values of that gene among all samples, which included PDXs and patients' samples. R version 4.1.0 (2021-05-18)" and "Matlab version 9.10.0.1684407 (R2021a) Update 3 were used.

**Reporting summary.** Further information on research design is available in the Nature Portfolio Reporting Summary linked to this article.

## Data availability

All relevant data generated or analyzed for this study are available within the paper and in Supplementary Information file or from the corresponding author upon reasonable request. Sequencing data and basic clinical annotations from all patients and PDX have been deposited in European Genome-phenome Archive (EGA; hosted by the EBI and CRG with the data set accession code EGAS00001005935 and EGAS00001007327 respectively. Further information about EGA can be found on https://ega-archive.org

("The European Genome-phenome Archive of human data consented for biomedical research"; http://www.nature.com/ng/journal/v47/n7/full/ng.3312.html). Source data for the graphs and charts in the main figures is available as Supplementary Data 3.

## Code availability

The analysis codes are available on a public GitHub server. The codes for HLA, metabolic and RNAseq analysis are available on: https://github.com/LimWChing/MAPPYACTS/blob/ac8191f0bea477309415ec54ccd6b920fdaa4915/Codes_Figure5.R.; https://github.com/cherkaos/PDXBiobankAnalysis and https://github.com/Rdroit/MAPPYACTs_PDX, respectively.

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

## Acknowledgements
We are highly grateful to all patients and parents that participated in the trial and consented to the development of preclinicssal models. We thank Noemie Assoun, Hélène Zakia Aid, Cécile Lopez, Karine Ser-Le Roux, Patrick Gonin and the animal facilities at GR, Gabriela Guillén, Ainara Magdaleno, Soledad Gallego at VHIR, pediatric surgeons, pathologists and oncologists from VH, Régis Brion, Frédéric Jacquot and Sarah Renault at the UTE core facilities at Nantes University; Olivia Bawa for technical assistance in histological and immunohistochemical techniques; Irène Villa for tissue section digitalization; Windy Rondof, Gerome Jules-Clement, Thierno Abdoulaye Balde for bioinformatics and data transfers; Tiphaine Adam-De-Beaumais, Imene Hezam, Eve Lapouble for operationals of MAPPYACTS. We are thankful to all investigators, surgeons, radiointerventionalists, pathologists and the treating clinical teams in the study centers and their referring hospitals: Gustave Roussy Cancer Campus (Villejuif, France), Institut Curie (Paris, France), Hôpital Trousseau (Paris, France), Hôpital Robert Debré (Paris, France), Hôpital Necker Enfants-Malades (Paris, France), Hôpital Sainte-Anne (Paris, France), Clinique Arago (Paris, France), Centre Marie Lannelongue (Plessis Robinsson, France), Hôpital Kremlin Bicêtre (Kremlin-Bîcètre), Institut d'hématologie et d'oncologie pédiatrique/Centre Léon Bérard/Hospice Civil de Lyon (Lyon, France), CHU La Timone (Marseile, France), CHU Toulouse (Toulouse, France), CHU Bordeaux (Bordeaux, France), CHU Nantes (Nantes, France) CHU Angers (Angers, France), CHU Strasbourg (Strasbourg, France), Centre Oscar Lambret (Lille, France), Fondazione IRCCS Istituto Nazionale dei Tumori (Milan, Italy), University Children's Hospital (Dublin, Ireland), Vall d'Hebron University Hospital (Barcelona, Spain). This work was supported by grants from Fondation Gustave Roussy; Fédération Enfants Cancers et Santé, Société Française de lutte contre les Cancers et les leucémies de l'Enfant et l'adolescent (SFCE), Association AREMIG and Thibault BRIET; Parrainage médecin-chercheur of Gustave Roussy; INSERM; Canceropôle Ile-de-France; Ligue Nationale Contre le Cancer (Equipe labellisée); Fondation ARC for the European projects ERA-NET on Translational Cancer Research (TRANSCAN 2) Joint Transnational Call 2014 (JTC 2014) 'Targeting Of Resistance in PEDiatric Oncology (TORPEDO)', ERA-NET TRANSCAN JTC 2014 (TRAN201501238), and TRANSCAN JTC 2017 (TRANS201801292); Agence Nationale de la Recherche (ANR-10-EQPX-03, Institut Curie Génomique d'Excellence (ICGex); IMI ITCC-P4; The Child Cancer Research Foundation (CCRF), Cancer Council Western Australia (CCWA); PAIR-Pédiatrie/CONECT-AML (INCa-ARC-LIGUE_11905 and Association Laurette Fugain), Ligue contre le cancer (Equipe labellisée, since 2016), OPALE Carnot institute; Dell; Fondation Bristol-Myers Squibb; Association Imagine for Margo; Association Manon Hope; L'Etoile de Martin; La Course de l'Espoir; M la vie avec Lisa; ADAM; Couleur Jade; Dans les pas du Géant; Courir pour Mathieu; Marabout de Ficelle; Olivier Chape; Les Bagouz à Manon; Association Hubert Gouin Enfance et Cancer; Les Amis de Claire; Kurt-und Senta Hermann Stiftung; Holcim Stiftung Wissen; Gertrud-Hagmann-Stiftung für Malignom-Forschung; Heidi Ras Grant Forschungszentrum fürs Kind; Children's Liver Tumor European Research Network (ChiLTERN) EU H2020 projet (668596); Fundación FERO and the Rotary Clubs Barcelona Eixample, Barcelona Diagonal, Santa Coloma de Gramanet, München-Blutenburg, Sassella-Stiftung, Berger-Janser Stiftung and Krebsliga Zürich, Deutschland Gemeindienst e.V. and others from Barcelona and province, and No Limits Contra el Cáncer Infantil Association.

## Author contributions
Conceptualization: M.E.M.C., A.M., J.S., S.Cherkaoui, R.J.M., S.M., T.M., F.P., E.D-D., D.S., B.G. Resources: R.J.M., S.M., T.M., S.T-E., I.G., F.P., J.C., F.R., N.E-W., A.S., A.V., S.Cairo., P.C., M.M., C.O., M.C., R.H-A., P.B., E.D-D., L.Z., L.L., G.P., O.D., G.S., D.S., B.G. Data curation: M.E.M.C., R.D., W.C.L., P.B., P.D., L.L., G.P., D.S., B.G. Software: R.D., W.C.L., A.M., J.S., S.Cherkaoui, R.J.M., T.M., P.D. Formal analysis: M.E.M.C., J-Y.S., R.D., W.C.L., A.M., J.S., S.Cherkaoui., R.J.M., S.M., T.M., P.D., D.S., B.G. Supervision: A.M., J.S., R.J.M., S.M., T.M., S.T-E., I.G., F.P., F.R., N.E-W., A.S., S.Cairo, P.C., C.O., M.C., R.H-A., E.D-D., L.Z., O.D., G.S., D.S., B.G. Investigation: M.E.M.C., S.Z., J-Y.S., R.D., W.C.L., A.M., J.S., S.Cherkaoui, R.J.M., A.L., S.M., T.M., S.T-E., I.G., F.P., J.C., F.R., N.E-W., A.S., A.V., S.Cairo, P.C., M.M., M.C., R.H-A., P.B., E.D-D., L.L., G.P., G.S., D.S., B.G. Methodology: M.E.M.C., S.Z., J-Y.S., R.D., W.C.L., A.M., J.S., S.C., R.J.M., A.L., S.M., T.M., F.P., J.C., E.D-D., B.G. Writing-original draft: M.E.M.C., A.M., J.S., S.C., R.J.M., S.M., D.S., B.G. Project administration: B.G. Writing-review and editing: M.E.M.C., S.Z., J-Y.S., R.D., W.C.L., A.M., J.S., S.C., R.J.M., A.L., S.M., T.M., S.T-E., I.G., F.P., J.C., F.R., N.E-W., A.S., A.V., S.C., P.C., M.M., C.O., M.C., R.H-A., P.B., E.D-D., P.D., L.Z., L.L., G.P., O.D., G.S., D.S., B.G.

## Competing interests
The authors declare no competing interests.

## Additional information

[1]INSERM U1015, Gustave Roussy Cancer Campus, Université Paris-Saclay, Villejuif, France. [2]Department of Pediatric and Adolescent Oncology, Gustave Roussy Cancer Campus, Villejuif, France. [3]INSERM U830, Equipe Labellisée LNCC, Diversity and Plasticity of Childhood Tumors Lab, PSL Research University, SIREDO Oncology Centre, Institut Curie Research Centre, Paris, France. [4]Department of Pathology and Laboratory Medicine, Translational Research Laboratory and Biobank, AMMICA, INSERM US23/CNRS UMS3655, Gustave Roussy Cancer Campus, Université

Paris-Saclay, Villejuif, France. [5]Gustave Roussy Cancer Campus, Bioinformatics Platform, AMMICA, INSERM US23/CNRS, UAR3655 Villejuif, France. [6]School of Data Sciences, Perdana University, Kuala Lumpur, Malaysia. [7]Division of Oncology and Children's Research Center, University Children's Hospital Zurich, University of Zurich, Zurich, Switzerland. [8]Gustave Roussy Cancer Campus, INSERM U1170, Université Paris-Saclay, Equipe labellisée Ligue Nationale Contre le Cancer, PEDIAC program, Villejuif, France. [9]Telethon Kids Institute – Cancer Centre, Perth Children's Hospital, Nedlands, WA, Australia. [10]Biological Ressources Center, Centre Léon Bérard, Lyon, France. [11]Small Animal Platform, Cancer Research Center of Lyon, INSERM U1052, CNRS UMR 5286, Centre Léon Bérard, Claude Bernard Université Lyon 1, Lyon, France. [12]UMR-E008 Stabilité Génétique, Cellules Souches et Radiations, Commissariat à l'Energie Atomique et aux Energies Alternatives (CEA), Université de Paris-Université Paris-Saclay, 92260 Fontenay-aux-Roses, France. [13]INSERM UMR 1238, Université Nantes, Nantes, France. [14]Pediatric Onco-Hematology Unit, University Hospital of Strasbourg, Strasbourg, UMR CNRS 7021, team tumoral signaling and therapeutic targets, University of Strasbourg, Faculty of Pharmacy, Illkirch, France. [15]Vall d'Hebron Research Institute (VHIR), Childhood Cancer and Blood Disorders Research Group, Division of Pediatric Hematology and Oncology, Vall d'Hebron Barcelona Hospital Campus, Barcelona, Spain. [16]Chemoresistance and Predictive Factors Group, Program Against Cancer Therapeutic Resistance (ProCURE), Catalan Institute of Oncology (ICO), Oncobell Program, Bellvitge Biomedical Research Institute (IDIBELL), L'Hospitalet del Llobregat, Xenopat SL, Parc Cientific de Barcelona (PCB), Barcelona, Spain. [17]XenTech, Evry, France. [18]Children University Hospital, Vandoeuvre-lès-Nancy, University of Nancy, Nancy, France. [19]Fondazione IRCCS Istituto Nazionale dei Tumori, Milan, Italy. [20]Paediatric Haematology/Oncology, Children's Health Ireland, Crumlin, Dublin, Republic of Ireland. [21]Unité de Génétique Somatique, Service d'oncogénétique, Institut Curie, Paris, France. [22]SiRIC RTOP (Recherche Translationnelle en Oncologie Pédiatrique); Translational Research Department, Institut Curie Research Center, PSL Research University, Institut Curie, Paris, France. [23]Present address: Balgrist University Hospital, University of Zurich, Zurich, Switzerland. [24]These authors jointly supervised this work: Didier Surdez, Birgit Geoerger. ✉email: birgit.geoerger@gustaveroussy.fr

