## [Peer Review File · Communications Biology]

Reviewers' comments:

Reviewer #1 (Remarks to the Author):

The paper presents 131 new pediatric PDX models, including 76 sarcoma, 26 non-CNS, 12 CNS, 3 lymphomas, 15 leukemias. This is a significant addition to the PDX repertoire for childhood cancer models and will be important to improvement of pediatric cancer testing. Generation of these models is a major endeavour and the contributing teams have done this in a systematic, rigorous fashion.

There are several notable strengths of the work. These include the HLA typing, the metabolic profiling, and the histological analysis. I have no major concerns about the work.

I would encourage the authors to improve their description of data and model sharing, as these aspects are crucial to the utility of the new PDX models. Supp Table S2 is an important part of this, but in the long run the sharing of other information such as sequencing data, metabolomic profiling, and histopathology images would be of significant value to the community. It is, of course, laborious to provide all this information, so I defer to the authors on how much of this they can provide. An expanded explanation of what might constitute a "Reasonable request," as well as a mention of what public databases the data will eventually reach, would be quite beneficial though. In particular, I would recommend the authors share descriptions of their models through public repositories such as PDCM Finder, which aggregate basic information about cancer models while providing contact links to the groups that have generated the models.

I have a few minor concerns noted below.

Minor concerns

p.17: "PDX models exhibited high consistency of the genetic alterations found in the patient samples."

-Please provide quantitative statistics on the fraction of genetic alterations that are preserved. On a related note, the methods statement that "Data were processed by bioinformatics analyses" needs clarification. The pipeline for mutation calling (e.g. mutation calling algorithm, sequencing depth, thresholds for variant calling, etc.) should be described.

Supp Fig. S3: "Microenvironment Principal Components (MPCs) in patient and PDX models (D) suggesting a partial reconstitution of osteosarcoma microenvironment in PDX, driven by the MPC1 component."

-It is unclear what in the figure supports the idea of partial reconstitution of the osteosarcoma microenvironment. There is no information of which patient samples correspond to which PDX samples. There is no mention in the figure of which samples are osteosarcomas.

p.18: "This clustering of the osteosarcoma microenvironment fraction in mice and patients is driven by the MPC1 (supplementary Fig.S3C)".

-There is no mention of osteosarcoma in Fig S3C. It simply shows 5 word clouds with no labeling of PDX samples, patient samples, or what "pos. cont. gene" and "neg. cont. gene" mean.

Reviewer #2 (Remarks to the Author):

Patient derived xenografts (PDX) can be a useful tool to generate patient avatars for in vivo and ex vivo drug testing, retrospective analysis of disease mechanisms and deepened molecular characterization when patient material is limited. Yet, reports that systematically assess PDX vs patient samples and the replicability of human cancers in the mouse remain scarce. In their study, Marques Da Costa et al. describe the systematic generation of PDX from different pediatric cancer entities associated with the MAPPYACTS trial and compare the generated models to the human disease. The study is well written and underlaid with comprehensive figures. I have some suggestions and questions to the authors to potentially improve the manuscript.

- 1) Please state how the PDX in the different entities can really be used to inform personalized medicine considering turnaround time from patient to PDX vs time to relapse. Can the samples help to drive p.m. or are they rather a retrospective tool? Do you have recommendations which patients to include/exclude for PDX generation? It would be interesting to hear if any treatments were informed based on the generated PDX avatars.
- 2) Please revise the color/shape coding of Figure 2. One can hardly distinguish the shown curves.
- 3) The authors report a high concordance of molecular characteristics of patient samples vs PDX-cohort. Yet, according to Fig 4A, this concordance varies strongly between different tumor entities. E. g. The T-ALL samples appear less conserved between human and PDX (e. g. IKFZ mutations patient samples but not PDX and vice versa). The authors should comment on the faithfulness of the entity-specific PDX models in more detail.
- 4) Concerning the ALL samples, the author report CD34 and CD38 which are expressed in both, B-ALL and T-ALL. However, critical markers in B-ALL (in terms of immunotherapy) are CD19, CD22. It would be interesting to see the expression of these molecules (or other other relevant targets) in patient vs PDX.
- 5) Different mouse strains were used for the amplification of tumor material. The choice of mouse strains is critical for successful engraftment.
 - a. Please state which mouse model was used for which entity and give a recommendation or better a detailed protocol per entity.
 - b. Have immunocompetent mice been used so that immunotherapy options can be tested?
- 6) Do CNS-tumor show CNS tropism in the applied mouse models when transplanted outside the CNS?
- 7) line 451 – Word missing: “In treatment remains the second [...] cause of cancer-related deaths.”

Reviewer #3 (Remarks to the Author):

The manuscript by Da Costa et al nicely describes the generation of multi-panel pediatric tumor PDX models and their molecular characterization with the focus on HLA class I genotype and metabolic profilings. As the lowest EFS is associated with relapsed and refractory diseases, the value of the presented work is in the development of a biobank of PDX models from relapsed/refractory patients. Furthermore, the new PDX models is a result of a coordinated work of several European research laboratories demonstrating the comprehensive multi-center academic-industry partnership. Of total 505 cancer samples, the 131 samples were established. Multiple passages (P1-P4) are collected for each model. Authors analyzed the HLA genotype status for generated PDX models which could be useful toward novel immune therapy approaches. Similarly, metabolic signature analysis allows evaluation of metabolic inhibitors. Additionally, each tumor panel was profiled for known genomic dysregulations. Overall, this is a valuable report of the unique resource potentially available to the preclinical research community. The manuscript is nicely written.

There are a few (rather minor) deficiencies/questions that could be addressed:

- Was the H&E staining performed in one laboratory, in a centralized manner? If no then the 12 discordant cases could be a result of differences in technical processing. Are PDXs expected to match patient tumor histology in the context of mouse stroma growth/contamination? It would be informative if some discussion was dedicated to the reasoning of the morphological analyses discordances.
- Discordant samples were excluded from sequencing for molecular characterization, while potentially it could be helpful for understanding whether the acquired genomic/transcriptomic changes are responsible for the observed discordance.
- The leukemia model establishment method is not fully described. It says that 'the model was considered established after the first growth in mice', but the description lacks the endpoint criteria of model establishment, the procedure, measurements, etc., as leukemia PDX engraftment approach differs from the methods for solid tumors PDX generation.
- Figure 6 lacks color/shape code for panel B.
- Is there a plan to report similar data/analyses for the 302 tumors that were transplanted but not yet engrafted?

Referee expertise:

Referee #1: computational, cancer genomics, evolution

Referee #2: cancer, models

Referee #3: therapeutics, cancer, models, pre-clinical testing

Reviewers' comments:

Reviewer #1 (Remarks to the Author):

The paper presents 131 new pediatric PDX models, including 76 sarcoma, 26 non-CNS, 12 CNS, 3 lymphomas, 15 leukemias. This a significant addition to the PDX repertoire for childhood cancer models and will be important to improvement of pediatric cancer testing. Generation of these models is a major endeavor and the contributing teams have done this in a systematic, rigorous fashion. There are several notable strengths of the work. These include the HLA typing, the metabolic profiling, and the histological analysis. I have no major concerns about the work.

I would encourage the authors to improve their description of data and model sharing, as these aspects are crucial to the utility of the new PDX models. Supp Table S2 is an important part of this, but in the long run the sharing of other information such as sequencing data, metabolomic profiling, and histopathology images would be of significant value to the community. It is, of course, laborious to provide all this information, so I defer to the authors on how much of this they can provide. An expanded explanation of what might constitute a “Reasonable request,” as well as a mention of what public databases the data will eventually reach, would be quite beneficial though. In particular, I would recommend the authors share descriptions of their models through public repositories such as PDCM Finder, which aggregate basic information about cancer models while providing contact links to the groups that have generated the model

R: We absolutely agree with the reviewer that data sharing is a crucial aspect for future research and we are happy to provide the sequencing data of the PDX as well as those of the originating patients that are available through the EGA number EGAS00001007327 and EGAS00001005935, respectively.

We have updated the Data sharing statement to: “Sequencing data and basic clinical annotations from all patients and PDX have been deposited in European Genome-phenome Archive (EGA; hosted by the EBI and CRG) with the data set accession code EGAS00001005935 and EGAS00001007327 respectively. Further information about EGA can be found on <https://ega-archive.org> (“The European Genome-phenome Archive of human data consented for biomedical research”; <http://www.nature.com/ng/journal/v47/n7/full/ng.3312.html>).”

I have a few minor concerns noted below.

Minor concerns

p.17: “PDX models exhibited high consistency of the genetic alterations found in the patient samples.”

-Please provide quantitative statistics on the fraction of genetic alterations that are preserved. On a related note, the methods statement that “Data were processed by bioinformatics analyses” needs clarification. The pipeline for mutation calling (e.g. mutation calling algorithm, sequencing depth, thresholds for variant calling, etc.) should be described.

R: In order to provide a complete overview of the conservation of somatic alterations and CNAs, we calculated the Jaccard distance between alterations shared by PDX and matched patients. To do so we used the full set of alterations without selection of putative driving events. We added this figure in the Figure S3 as A and B and stated in the manuscript on the observed variations that is highly biased by the sampling and the tumor purity (included in page 18):

“Biased by the sampling and the tumor purity, the quantification of the alteration conservation using the Jaccard distance between matched patient and PDX suggests that osteosarcomas and rhabdomyosarcomas are the most divergent, in accordance with their unstable genetics.”

We have clarified the mutation calling as followed on page 40. We added the following in the manuscript (Material and methods page 43): The variants are annotated with Annovar. Somatic alterations with less than 5 minimum reads supporting the mutations, 5% of the reads covering the sequence supporting the alteration and more than 1% of the population annotated with the mutation in the databases 1000g2015aug (latest 1000 Genomes Project dataset with allele frequencies in six populations including ALL, African, Admixed American, East Asian, European and South Asian) and kaviar_20150923 (latest Kaviar database with 170 million variants from 13K genomes and 64K exomes) were filtered out. Samples similarity estimation based on the somatic mutations was performed using Jaccard distance.

Supp Fig. S3: “Microenvironment Principal Components (MPCs) in patient and PDX models (D) suggesting a partial reconstitution of osteosarcoma microenvironment in PDX, driven by the MPC1 component.”-It is unclear what in the figure supports the idea of partial reconstitution of the osteosarcoma microenvironment. There is no information of which patient samples correspond to which PDX samples. There is no mention in the figure of which samples are osteosarcomas.

R: There are two osteosarcoma PDX models with a high negative MPC1 contribution similar to the osteosarcoma patient tumors. We added a figure S3F, illustrating the presence of these two models with a similar ME as the osteosarcoma patient tumors. The legend of supplementary figure S3 has been modified accordingly:

“Microenvironment Principal Components (MPCs) in patient and PDX models (G) suggesting a partial reconstitution of osteosarcoma microenvironment in 2 PDX, driven by the MPC1 component (F).”

p.18: “This clustering of the osteosarcoma microenvironment fraction in mice and patients is driven by the MPC1 (supplementary Fig.S3C)”. -There is no mention of osteosarcoma in Fig S3C. It simply shows 5 word clouds with no labeling of PDX samples, patient samples, or what “pos. cont. gene” and “neg. cont. gene” mean.

R: We thank the reviewer for pointing this out and have clarified it to the reader with the following modification: “This clustering of the osteosarcoma microenvironment fraction in mice and patients is driven by the MPC1 (revised supplementary Fig.S3F)”.

We modified accordingly the legend of the figure to clarify the meaning of “pos. cont. gene” and “neg. cont. gene”. Sign of the contribution illustrates if the sample participates to the negative or positive functional enrichment fraction of the corresponding principal component.

Reviewer #2 (Remarks to the Author):

Patient derived xenografts (PDX) can be useful tool to generate patient avatars for in vivo and ex vivo drug testing, retrospective analysis of disease mechanisms and deepened molecular characterization when patient material is limited. Yet, reports that systematically assess PDX vs patient samples and the replicability of human cancers in the mouse remain scarce. In their study, Marques Da Costa et al. describe the systematic generation of PDX from different pediatric cancer entities associated with the MAPPYACTS trial and compare the generated models to the human disease. The study is well written and underlaid with comprehensive figures. I have some suggestions and questions to the authors to potentially improve the manuscript.

1) Please state how the PDX in the different entities can really be used to inform personalized medicine considering turnaround time from patient to PDX vs time to relapse. Can the samples help to drive p.m. or are they rather a retrospective tool? Do you have recommendations which patients to include/exclude for PDX generation? It would be interesting to hear if any treatments were informed based on the generated PDX avatars.

R: We thank the reviewer for this highly relevant comment. Indeed, so far, our main purpose was to establish models from recurrent or refractory patients that are the population of most medical need in pediatric oncology and that need new treatment strategies. Indeed, in the last decade we use a precision cancer medicine approach for advanced patients where molecular profiling aims to guide further salvage therapies adapted to the biological findings (*Berlanga et al. Cancer Discovery 2022*). There are indeed several efforts to provide additional functional information on tumor sensitivity to anticancer agents, most of them in vitro and recently increasingly also in organoid models. As the reviewer highlights, PDX are limited in their use as the tumor take rate and delay for established models are limiting factors. They are nevertheless still considered as closer to the patient than current cultures avatars. Most of our models have been provided to the ITCC-P4 consortium and will thus be available for high throughput evaluations. It was important to us to provide the most detailed characterization of the corresponding patient to allow researcher to explore mechanism of action studies as most as possible.

For the purpose of this publication, we have not provided testing studies, as they were in part performed in specific studies as mentioned in the discussion (“Several of our models already

have been used for preclinical explorations and were included in publications demonstrating the proof-of-concept for their relevance. Among those were Burkitt lymphoma ⁴⁰, anaplastic large cell lymphoma ⁴¹, neuroblastoma ⁴²⁻⁴⁴, Ewing sarcoma ⁴⁵, rhabdomyosarcoma ^{46,47} and acute lymphatic leukemia ⁴⁸ PDX models”) and we felt that they are out of the scope of this publication where we focused on the comprehensive characterization.

2) Please revise the color/shape coding of Figure 2. One can hardly distinguish the shown curves

R: We thank the reviewer for this suggestion for figure 2. We modified the graphs, and colored curves have now been used to improve visibility. The legend was also modified in order to include these modifications.

3) The authors report a high concordance of molecular characteristics of patient samples vs PDX-cohort. Yet, according to Fig 4A, this concordance varies strongly between different tumor entities. E. g. The T-ALL samples appear less conserved between human and PDX (e. g. IKZF mutations patient samples but not PDX and vice versa). The authors should comment on the faithfulness of the entity-specific PDX models in more detail.

R: We thank the reviewer for this comment and indeed some models looked less consistent when plotting only selected genes that were retained by the molecular tumor board or disease specific, as chosen for this oncoplot. Concerning for example T-ALL, the CEA group who developed the T-ALL PDX models have previously published that clonal selection was observed in T-ALL PDX models, and this could be related to the speed of leukemia development in immune-deficient mice (Poglio et al, Oncotarget, 2016). Moreover, the same group also showed that the clonal selection observed in T-ALL PDX could fit with enhanced aggressiveness of the selected cells (Clappier, JEM, 2010). Here, as outlined by the reviewer, a *IKZF1* mutation could be revealed in the selected PDX cells from TALL-3, and such event has been related to enhanced Notch signaling (Dumortier A et al, MCB, 2006), a very commonly activated pathway in T-ALL. Interestingly, in the same sample, the *NOTCH1* mutation present in the patient cells was lost in the PDX cells, supporting that the *IKZF1* mutation may substitute for this loss in terms of NOTCH pathway activation.

In reply to also this reviewer's question above on the consistency, we now provided in addition a complete overview of the conservation of somatic alterations and CNAs, calculated as the Jaccard distance between alterations shared by PDX and matched patients. For this, we used the full set of alterations without selection of putative driving events. We added this figure in the Figure S3 as A and B and stated in the manuscript on the observed variations that is highly biased by the sampling and the tumor purity (included in page 18):

"Biased by the sampling and the tumor purity, the quantification of the alteration conservation using the Jaccard distance between matched patient and PDX suggests that osteosarcomas and rhabdomyosarcomas are the most divergent, in accordance with their unstable genetics."

4) Concerning the ALL samples, the author report CD34 and CD38 which are expressed in both, B-ALL and T-ALL. However, critical markers in B-ALL (in terms of immunotherapy) are CD19, CD22. It would be interesting to see the expression of these molecules (or other relevant targets) in patient vs PDX.

R: We agree with the reviewer that this is an interesting point and appreciate the opportunity to clarify this point. As mentioned in the legend of figure S6, all dot plots showing expression of CD34 and CD38 are gated on human CD19. Indeed, for these cases, all blasts cells were CD19+. On top of CD19, CD34 and CD38, the B-ALL flow cytometry panel was also composed of CD10, human CD45, and murine CD45.1 (to exclude murine cells at the analysis) but unfortunately did not contain CD22.

5) Different mouse strains were used for the amplification of tumor material. The choice of mouse strains is critical for successful engraftment.

a. Please state which mouse model was used for which entity and give a recommendation or better a detailed protocol per entity.

R: We agree that different mouse strains may influence tumor engraftment and growth. Table S2 mentions for each model the mouse strain used at P0 and P2, as well as the implantation method. For the text we have opted to list in Material and Methods all mouse strains and implantation methods that were used in the program (page 37): “For solid tumors, heterotopic and/or orthotopic PDX were established depending on the tumor size and tumor histology in 3-7 weeks immunocompromised female and male Swiss athymic Nude (CrI:NU(Ico)-Foxn1nu), SCID (CB17/Icr-Prkdcscid/IcrIcoCrI), NSG (NOD.Cg-PrkdcscidIL2rgtm1Wjl/SzJ) or NSG expressing human cytokines (NSG-IL) mice, obtained from the institutional animal facilities or from Charles River animal facilities, as reported previously ⁴⁹⁻⁵². Tumor samples of 2-5 mm³ were implanted subcutaneously (SC) on one or both flanks, in the sub interscapular fat pad (FP), paratibially with periosteum activation (PT) ⁵² (for bone tumors), intramuscular (IM) (rhabdomyosarcoma), or in the left kidney capsule (KC) (samples smaller than 2mm³) into 2-5 mice. For central nervous system (CNS) tumors, tumor samples were digested and cell homogenates injected intracerebral into the caudate nucleate, the cerebellum or the pons (IC) of nude mice. For leukemia samples, mononuclear cells (50,000 to 10⁶ cells) were either directly injected into the femur or intravenously transplanted in sub-lethally irradiated (2.5 Gy) NSG mice. “

We reported the tumor engraftment and take rate per histology type that show clear differences as discussed. Nevertheless, also the expertise of the team may influence the engraftment rate and as 10 different laboratories were involved, we did not feel comfortable to overinterpret this aspect and give strong recommendations.

b. Have immunocompetent mice been used so that immunotherapy options can be tested?

R: So far, no immunocompetent mice have been used but this is planned in the ITCC-P4 project as indeed the question of immune response is currently an important question and debate.

6) Do CNS-tumor show CNS tropism in the applied mouse models when transplanted outside the CNS?

R: We have not explored this especially and have not observed in the subcutaneous CNS tumor PDX any tropism to the CNS.

7) line 451 – Word missing: “In treatment remains the second [...] cause of cancer-related deaths.”

R: We apologize for the error and corrected the main text: “Leukemia is the most common type of cancer in children and, despite significant advances in treatment, it remains the second cause of cancer-related deaths.”

Reviewer #3 (Remarks to the Author):

The manuscript by Da Costa et al nicely describes the generation of multi-panel pediatric tumor PDX models and their molecular characterization with the focus on HLA class I genotype and metabolic profiling. As the lowest EFS is associated with relapsed and refractory diseases, the value of the presented work is in the development of a biobank of PDX models from relapsed/refractory patients. Furthermore, the new PDX models is a result of a coordinated work

of several European research laboratories demonstrating the comprehensive multi-center academic-industry partnership. Of total 505 cancer samples, the 131 samples were established. Multiple passages (P1-P4) are collected for each model. Authors analyzed the HLA genotype status for generated PDX models which could be useful toward novel immune therapy approaches. Similarly, metabolic signature analysis allows evaluation of metabolic inhibitors. Additionally, each tumor panel was profiled for known genomic dysregulations. Overall, this is a valuable report of the unique resource potentially available to the preclinical research community. The manuscript is nicely written.

There are a few (rather minor) deficiencies/questions that could be addressed:
- Was the H&E staining performed in one laboratory, in a centralized manner? If no then the 12 discordant cases could be a result of differences in technical processing. Are PDXs expected to match patient tumor histology in the context of mouse stroma growth/contamination? It would be informative if some discussion was dedicated to the reasoning of the morphological analyses discordances.

R: H&E staining was performed in the same laboratory with the same standard procedures. The comparison between primary tumors and PDX was based on morphology alone; no complementary immunohistochemical stain was used. Discordances usually came from the absence in PDX samples of the suggestive morphological features present in the initial tumor and used for the pathological diagnosis; if this results from a process of “dedifferentiation” or from a sample bias is open to discussion. In other cases, both the initial tumor and its PDX were too poorly differentiated to allow a precise morphological diagnosis, without the help of ancillary techniques. The reasons for discordances were mentioned in the description of the cases and we clarified that it was done centrally in one laboratory and reviewed by one pathologist.

- Discordant samples were excluded from sequencing for molecular characterization, while potentially it could be helpful for understanding whether the acquired genomic/transcriptomic changes are responsible for the observed discordance.

R: The discordant samples were not excluded from sequencing. We did the investigation and the discordance was not observed at the genetic and transcriptomic levels.

- The leukemia model establishment method is not fully described. It says that 'the model was considered established after the first growth in mice', but the description lacks the endpoint criteria of model establishment, the procedure, measurements, etc., as leukemia PDX engraftment approach differs from the methods for solid tumors PDX generation.

R: Following injection of patients' cells into primary recipients (the injection route is indicated in Supplementary Table 2), bone marrow and blood samplings were performed and the human leukemic cell infiltration was measured by flow cytometry using the following markers (T-ALL: hCD45, hCD7; B-ALL: hCD45, hCD34, hCD38, hCD19 and hCD10; AML: hCD45, hCD34, hKIT, hCD33). The end points and protocols for each subtype of leukemia is indicated below.

For T-ALL: Bone marrow samplings (maximum 3/mouse) were done on human T-ALL mouse recipients and flow cytometry was used to measure the leukemic infiltration. When %hCD45+CD7+ leukemic cells >80%: mice were euthanatized and cells recovered from mouse femurs (not spleen). BM cells (5-10M/vial) were then frozen in SVF+10%DMSO in liquid nitrogen. Some cells were directly frozen in RLT+ buffer in order to perform RNAseq. Also, cell samples (1-5M/mouse) from infiltrated mouse BM were centrifuged in order to obtain dry pellets and perform mutation analyses. When no human leukemic cells were detected after 6 months, mice were considered as negative.

For B-ALL and AML: endpoint criteria include a primary engraftment within less than 6 months and a percentage of blasts positive for one of the indicated markers >20% in primary recipients. $1-3 \times 10^6$ cells from the bone marrow of primary recipients were then transplanted into secondary (P2) recipients. For all P2 models presented here, the percentage of human blasts in the bone marrow were >80% at endpoint. Note that several models showed discrepancies between the % of blasts in the bone marrow and the % of blasts in blood, the latter being frequently lower than in bone marrow. Therefore, the % of blasts in the bone marrow was retained as the main criteria. BM cells were frozen in SVF+10%DMSO, in RLT+ buffer or as dry pellet for subsequent analyses.

We added the following in the manuscript (Material and methods page 39):

For leukemia samples, mononuclear cells (50,000 to 10^6 cells) were either directly injected into the femur or intravenously transplanted in sub-lethally irradiated (2.5 Gy) NSG mice (the injection route is indicated in Supplementary Table 2). Following injection of patients' cells into primary recipients, bone marrow and blood samplings were performed and the human leukemic cell infiltration was measured by flow cytometry using the following markers (T-ALL: hCD45, hCD7; B-ALL: hCD45, hCD34, hCD38, hCD19 and hCD10; AML: hCD45, hCD34, hKIT, hCD33). For T-ALL: When %hCD45+CD7+ leukemic cells >80%: mice were euthanatized and cells recovered from mouse femurs. For B-ALL and AML: endpoint criteria include a primary engraftment within less than 6 months and a percentage of blasts positive for one of the indicated markers >20% in primary recipients. $1-3 \times 10^6$ cells from the bone marrow of primary recipients were then transplanted into secondary (P2) recipients. For P2, the % of blasts in the bone marrow was retained as the main criteria (>80% at endpoint).

- Figure 6 lacks color/shape code for panel B.

R: We thank the reviewer for his comment on the legend of Figure 6B. We now improved the visibility of the legend and display it at the beginning of Figure 6B.

- Is there a plan to report similar data/analyses for the 302 tumors that were transplanted but not yet engrafted?

R: We have not planned to provide a similar detailed description of the ones that could not be established, but indeed planned an analysis to see which factors could be involved in the successful establishment. Up to now, our attempts to identify significant differences between successful and failed engraftments at the Omics level didn't converge to significant results. This kind of study may be limited by the cohort size and the heterogeneity of the studied pathologies, leading to relatively low statistical power for such investigation. One project was to provide the information on patients for whom -frozen material is still available which may guide in selective models development in interesting cases.

REVIEWERS' COMMENTS:

Reviewer #1 (Remarks to the Author):

The authors have addressed my concerns.

Reviewer #2 (Remarks to the Author):

My suggestions were adequately considered and addressed. From my point of view, this manuscript is feasible for publication.

Reviewer #3 (Remarks to the Author):

This is a revised manuscript by Da Costa et al. Overall, the authors addressed the presented concerns, which were minor.